# Class A and B GPCRs trigger rapid Gα$_s$ translocation to late and slow recycling endosomes
Andréanne Laniel, Brian Holleran, Émy Labonté, Sarah-Janne Grondin, Pierre-Luc Boudreault [ID] & Christine Lavoie [ID] [✉]

Gα$_s$ is classically known for mediating G protein-coupled receptor (GPCR) signaling at the plasma membrane (PM), but it is now established that Gα$_s$ also supports a second wave of signaling from internalized GPCRs within early endosomes. However, the mechanisms underlying Gα$_s$ trafficking remain unclear. Here, using live-cell confocal microscopy and bioluminescence resonance energy transfer (BRET) assays, we investigated Gα$_s$-GFP dynamics following activation of class A (β$_2$AR) and class B (V$_2$R) receptors, which exhibit different level of endosomal signaling. Our findings demonstrate that Gα$_s$ rapidly (< 2 min) translocates to late (Rab7) and slow recycling (Rab11) endosomes, bypassing the classical endocytic route and displaying only transient colocalization with receptors. This trafficking depends on Gα$_s$ activation at the PM, its release from the membrane, and an intact palmitoylation site, but occurs independently of receptor internalization. This work shed light on non-canonical route for Gα$_s$ endosomal trafficking, with important implications for endosomal GPCR signaling.

The human genome encodes approximately 1000 G protein-coupled receptors (GPCRs), making them the largest family of cell surface receptors[1,2]. Upon agonist binding, a GPCR undergoes a conformational change, stabilizing it in an active form that promotes interaction with heterotrimeric G proteins (Gαβγ). This interaction facilitates the exchange of GDP for GTP on the Gα subunit, activating it and enabling the dissociation of Gα from the Gβγ dimer[2,3]. G proteins are central to transmitting signals initiated by extracellular stimuli[4]. The Gα subunits are divided into four families—Gα$_s$, Gα$_{i/o}$, Gα$_{q/11}$, and Gα$_{12/13}$—each linked to specific effectors and second messenger pathways[5,6]. For instance, Gα$_s$ activates adenylate cyclase (AC), leading to the production of cyclic adenosine monophosphate (cAMP)[2].

The canonical activation and signaling of GPCRs at the PM have been extensively characterized. However, it has become evident that GPCRs can also signal from intracellular compartments, including endosomes and the Golgi apparatus[7–11]. Many Gα$_s$-coupled GPCRs have been reported to initiate endosomal G protein activity by triggering a second round of agonist-dependent allosteric coupling on the endosomal membrane, leading to distinct signaling outcomes compared to those initiated at the PM[12–14]. Evidence supporting this model includes the necessity of receptor internalization for endosomal G protein signaling and the requirement for receptor activation on endosomal membranes[9,11,13,15–24]. The interaction between GPCRs and β-arrestins, proteins involved in GPCR desensitization and internalization, has also been reported to play a significant role in endosomal signaling. Some GPCRs, referred to as class A GPCRs, rapidly lose their interaction with β-arrestins, while others maintain a sustained interaction (known as class B GPCRs)[25,26]. Several class B GPCRs, including the parathyroid hormone receptor (PTHR), neurokinin 1 receptor (NK1R), and vasopressin type 2 receptor (V$_2$R), have been shown to continue signaling from endosome, rather than remaining desensitized[21,27,28].

While much is understood about the machinery responsible for GPCR internalization and their post-endocytosis trafficking—where some receptors are efficiently recycled back to the PM and others are directed toward degradation—less is known about the trafficking processes of G protein heterotrimers from the PM through various intracellular compartments[29,30]. Among G proteins, Gα$_s$ trafficking has been the most extensively studied. Initially, Gα$_s$ is anchored at the PM via palmitoylation[31,32]. Previous studies have shown that receptor activation leads to the dissociation of Gα$_s$ from the PM into the cytoplasm[33–35]. This depalmitoylation reaction is facilitated by enzymes such as acyl-protein thioesterases, allowing the cytoplasmic pool of Gα$_s$ to translocate to various subcellular compartments[36,37]. Imaging studies have indicated that internalized Gα$_s$ may appear diffuse in the cytosol, although association with intracellular vesicles has also been observed[33,34,38–40]. Some studies report that Gα$_s$ is present on intracellular vesicles that are distinct from those containing the internalized receptors, while others indicate colocalization between the two[11,13,23,39]. However, the mechanisms underlying the recruitment of Gα$_s$ to intracellular

Institut de Pharmacologie de Sherbrooke, Department of Pharmacology and Physiology, Faculty of Medicine and Health Sciences, Université de Sherbrooke, Sherbrooke, Quebec, QC, Canada. [✉]e-mail: Christine.L.Lavoie@USherbrooke.ca

compartments remain poorly understood. Repalmitoylation on intracellular membranes has been proposed as a potential mechanism, alongside recruitment by Gβγ through the internalization of a receptor-βarrestin-Gβγ complex[35,38,39,41]. Furthermore, while Gα$_s$ has been detected on endosomes, the specific nature of these endosomes and their trafficking itinerary remain poorly defined[37,39].

In this study, we utilized live-cell confocal microscopy and bioluminescence resonance energy transfer (BRET) assays to investigate Gα$_s$-GFP internalization following the stimulation of prototypical class A (β$_2$AR, D$_1$R) and class B (V$_2$R, PTHR) receptors. We examined how receptor activation and internalization influence this process using various agonists, as well as the nature of the endosomal compartments targeted by Gα$_s$. Our findings reveal that Gα$_s$ is rapidly translocated to vesicles within less than 2 minutes of receptor stimulation and displays only partial and transient colocalization with the receptors. Surprisingly, Gα$_s$ was rapidly and predominantly recruited to late endosomes (Rab7) and slow recycling endosomes (Rab11). Furthermore, Gα$_s$ endosomal recruitment was found to be independent of GPCR internalization but dependent of Gα$_s$ activation at the PM, its dissociation from the membrane, and its palmitoylation site. These findings have significant implications for our understanding of GPCR signaling from endosomes.

## Results

### Kinetics of Gα$_s$ internalization following stimulation of β$_2$AR and V$_2$R

To investigate the internalization of Gα$_s$ following GPCR activation, we examined the subcellular localization of GFP-labeled Gα$_s$ using live cell confocal microscopy. GFP was integrated into the α1/αA loop of Gα$_s$, a modification shown to preserve its functionality[39]. We co-expressed human Gα$_s$-GFP along with various cell surface SNAP-tagged GPCRs in HEK293 cells. To label the receptors at the PM, cells were preincubated with cell-impermeant fluorescent substrate SNAP-Surface 649. We first analyzed the behavior of Gα$_s$-GFP in response to stimulation of SNAP-β$_2$AR, a prototypical Gα$_s$-coupled class A GPCR. Cells were treated with isoproterenol (Iso) for 0, 2, 5, 7, and 10 min (Fig. 1A). Prior to stimulation, Gα$_s$ was primarily localized at the PM, consistent with its association with inactive β$_2$AR, but it also displayed some diffuse presence in the cytoplasm. Cell-to-cell variability was observed in the degree of cytoplasmic Gα$_s$-GFP signal, consistent with differences in transfection efficiency and expression levels. The apparent cytoplasmic localization is most likely attributable to over-expression, as the distribution of endogenous Gα$_s$ in live cells has not yet been established. To minimize this variability, only cells displaying clear plasma membrane localization were selected for imaging and quantitative analyses. After 2 minutes of stimulation with isoproterenol, Gα$_s$ began to appear on intracellular vesicles (Fig. 1A), with this vesicular localization becoming more pronounced and numerous over time. Notably, colocalization between Gα$_s$-GFP and internalized β$_2$AR was minimal; at all time points, only a few intracellular vesicles exhibited partial and transient overlap between the two signals, indicating that their association is both limited and short-lived.

To determine whether the size of the GFP tag affected Gα$_s$ trafficking, we employed an alternative strategy. Cells were transfected with Gα$_s$ modified at the position 77–81 to insert a short HA tag, along with a frankenbody-GFP, a genetically encoded antibody-based probe designed to detect the HA epitope[42,43]. Using these tools, we observed that the formation of punctate structures for Gα$_s$ occurred with kinetics identical to those seen with Gα$_s$-GFP following β$_2$AR stimulation with Iso (Supplementary Fig. 1). However, a limitation of this approach was the high level of GFP signal in the

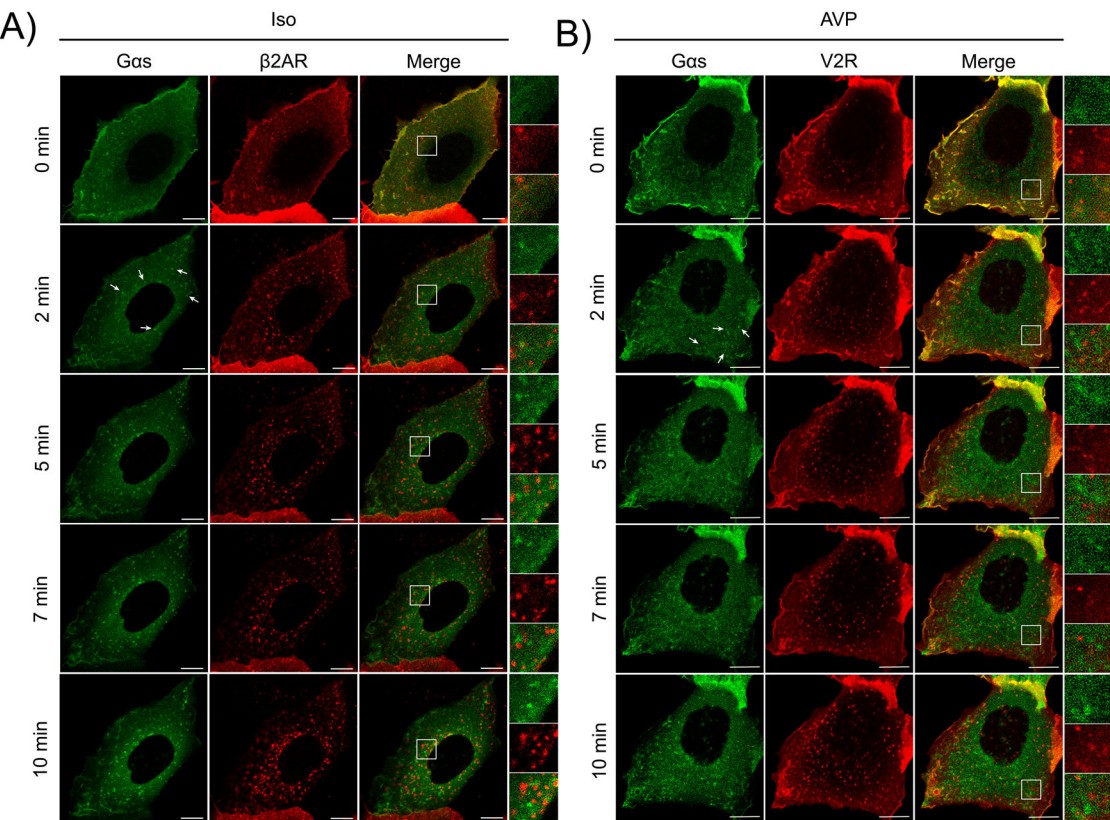

**Fig. 1 | Gα$_s$ is Internalized Following Stimulation of β$_2$AR and V$_2$R.** HEK293 cells expressing Gα$_s$-EGFP and SNAP-tagged GPCRs were starved and incubated at 37 °C for 20 min with SNAP-649 prior to a 10 min stimulation with agonists at 37 °C, followed by analysis using live-cell confocal microscopy. **A** SNAP-β$_2$AR was incubated with 1:1000 SNAP-649, followed by stimulation with 10 μM Isoproterenol (Iso). **B** SNAP-V$_2$R was incubated with 1:1000 SNAP-649, then stimulated with 100 nM AVP. White arrows indicate examples of Gα$_s$ vesicles that appears at 2 min. Scale bars, 10 μm.

**Fig. 2 | Endosomal Distribution of Gα_s Following β_2AR Stimulation.** Live-cell confocal microscopy was performed on starved HEK293 cells expressing Gα_s-EGFP and Flag-β_2AR, along with various endosomal markers: **A** mCherry-Rab4, **B** RFP-Rab5, **C** RFP-Rab7, and **D** mCherry-Rab11. Cells were stimulated with 10 μM isoproterenol (Iso) for a total of 20 min at 37 °C. Scale bars, 10 μm.

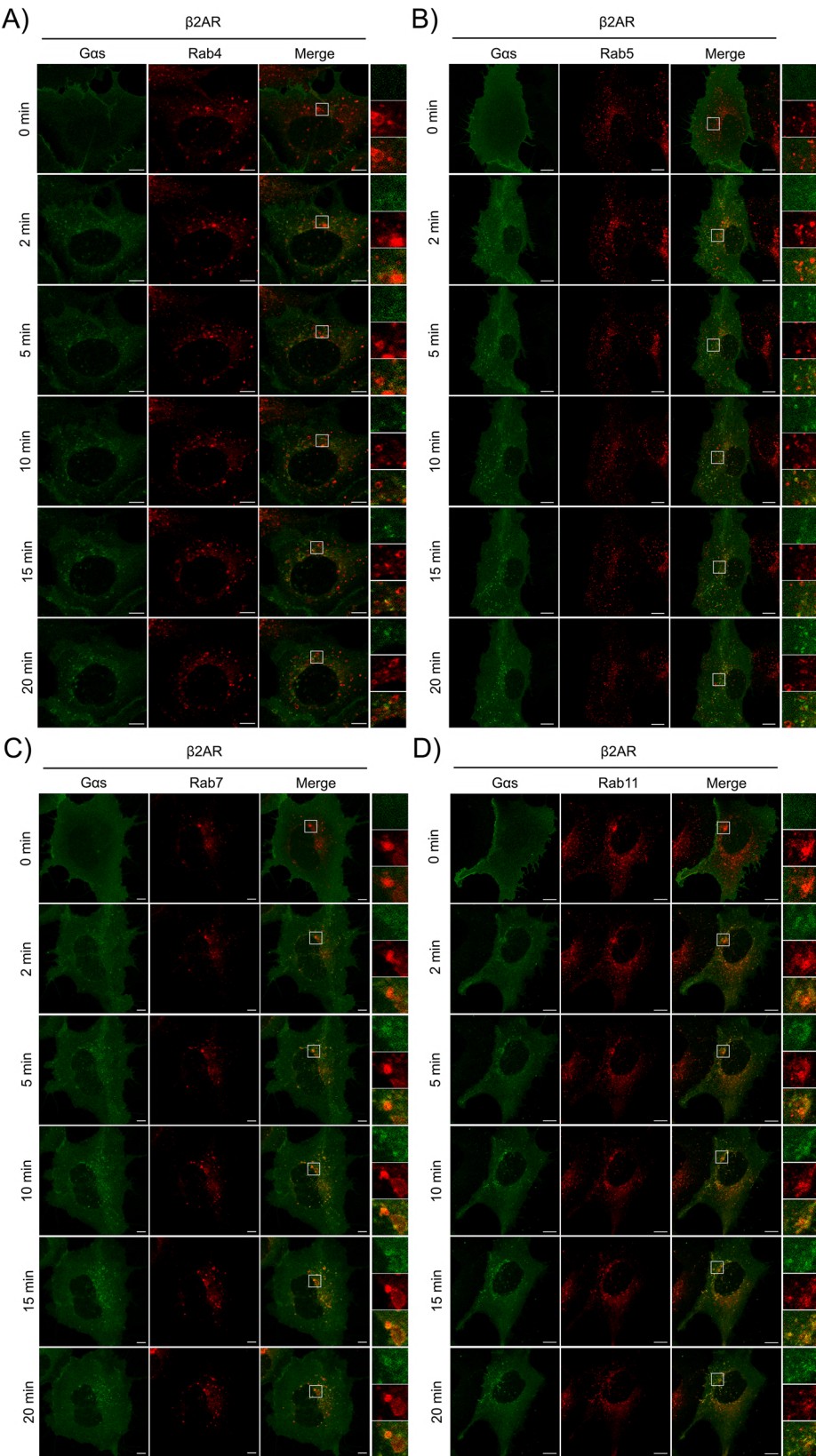

cytoplasm due to frankenbody-GFP expression. Moreover, although these findings suggest that tag size does not overtly affect Gα_s trafficking, it remains possible that even small tags introduce subtle effects on protein behavior; therefore, these constructs may not fully recapitulate wild-type Gα_s trafficking.

Next, we analyzed the subcellular distribution of Gα_s-GFP in cells expressing SNAP-tagged V_2R, a prototypical Gα_s-coupled class B GPCR known to sustain Gα_s signaling in endosome (Fig. 1B)[21]. Prior to stimulation with arginine-vasopressin (AVP), Gα_s-GFP was predominantly localized at the PM, with some diffuse cytoplasmic signal. Upon AVP treatment, Gα_s-

**Fig. 3 | Endosomal Distribution of Gα_s Following V_2R Stimulation.** Starved HEK293 expressing Gα_s-EGFP, Myc-V_2R and **A** mCherry-Rab4, **B** RFP-Rab5, **C** RFP-Rab7, **D** mCherry-Rab11 were stimulated with 100 nM AVP for a total of 20 min at 37 °C. Live-cell analysis was performed using confocal microscopy. Scale bars, 10 μm.

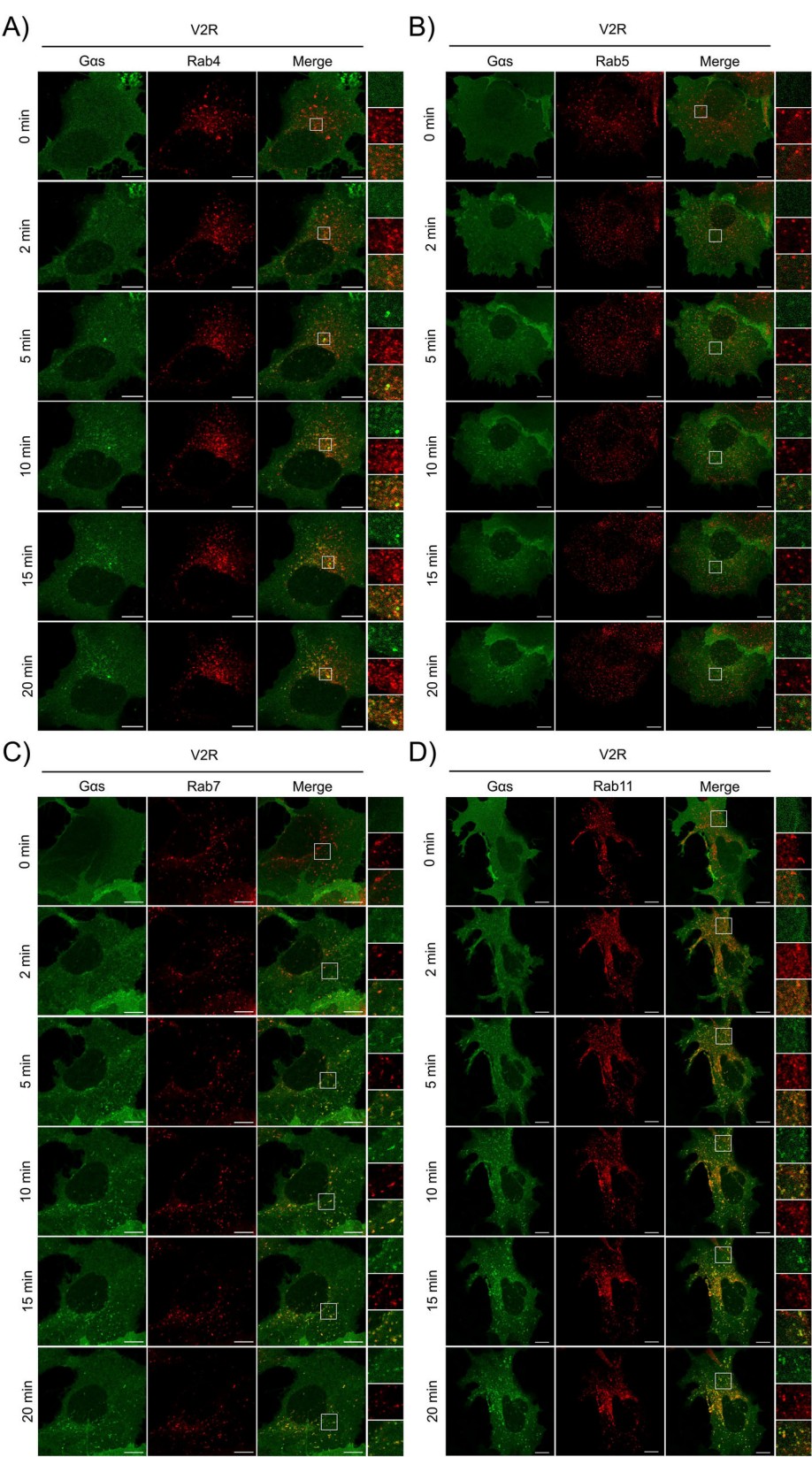

GFP redistributed to intracellular puncta within 2 minutes, and this vesicular localization became progressively more pronounced at later time points. Similar to the observations with β_2AR, only limited and transient colocalization was observed on a small subset of vesicles.

Together, these findings suggest that activation of Gα_s-coupled receptors (either class A or B) triggers a rapid translocation of Gα_s to intracellular vesicles as early as 2 minutes post-stimulation. However, this translocation is accompanied only by partial and transient colocalization

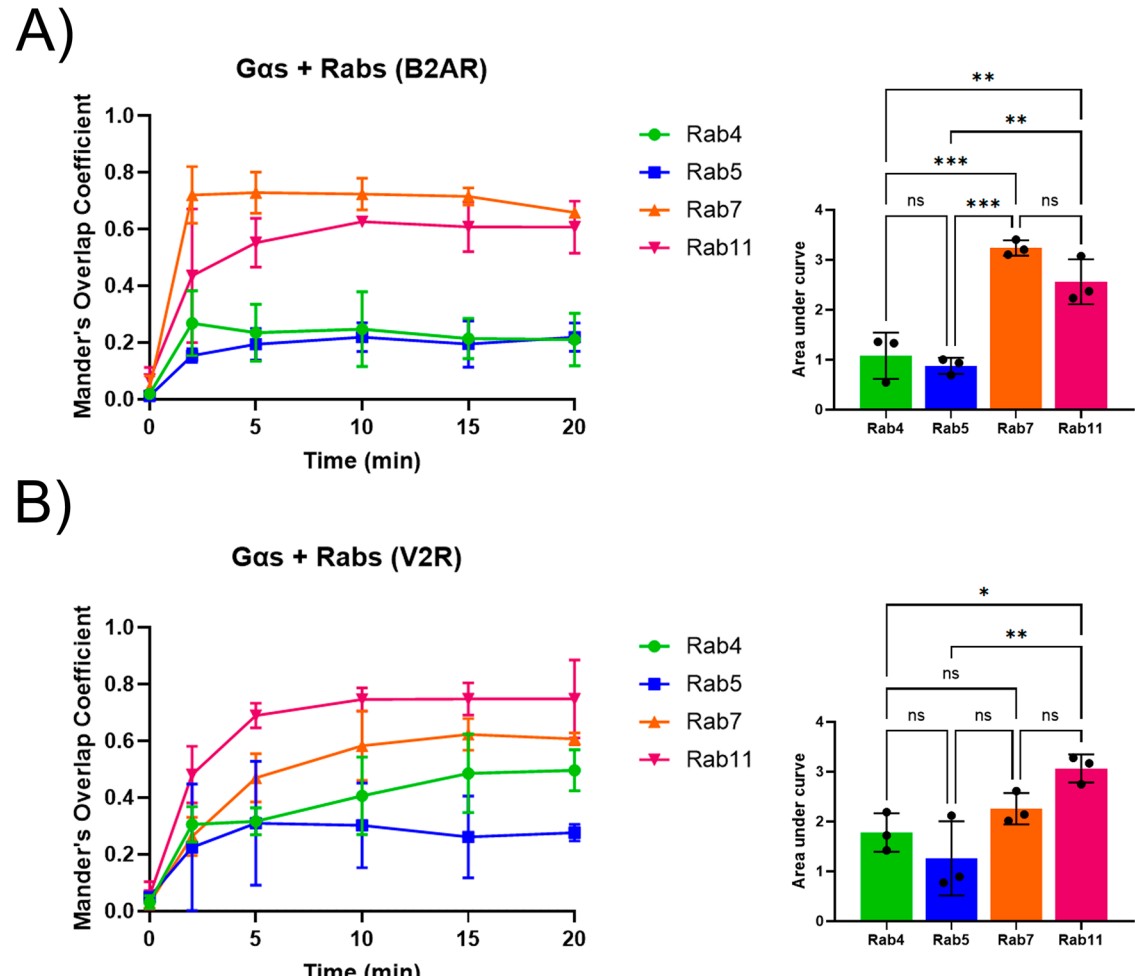

**Fig. 4 | Quantification of Colocalization Between Gα$_s$ and Rab Markers.** Mander's overlap coefficient (MOC) was used to quantify the colocalization of Gα$_s$ and Rab markers observed in Figs. 2 and 3. Error bars represent the mean ± S.D. of $n$ = 3 independent experiments. For each condition, 4 to 10 cells were quantified, with the same cells assessed over time (0–20 min post-stimulation). XY graphs illustrate the colocalization of Gα$_s$ with Rab4, Rab5, Rab7, and Rab11 following stimulation with **A** β$_2$AR (10 μM Iso) or **B** V$_2$R (100 nM AVP). Histograms display the area under the curve for each XY graph. Statistical analysis was conducted using one-way ANOVA with Tukey's multiple comparisons test, where * p < 0.05; ** p < 0.01; *** p < 0.001; ns indicates not significant.

with the internalized receptors, suggesting that Gα$_s$ and its cognate receptors largely follow distinct trafficking pathways following activation.

## Endosomal distribution of Gα$_s$ upon β$_2$AR and V$_2$R activation

Gα$_s$ has been previously reported to translocate to various intracellular compartments, primarily early endosomes, where it would mediate a secondary round of activation following coupling with internalized GPCRs[11,37,39]. However, the spatiotemporal organization of Gα$_s$ within specific compartments of the endosomal pathway has not been described. To explore the nature of the endosomes labelled by Gα$_s$ and the kinetic of its translocation on these specific endosomes, we compared the localization of Gα$_s$-GFP to a panel of Rab proteins, which serve as markers for distinct endosomal compartments: Rab4 (rapid recycling endosomes), Rab5 (early endosomes), Rab7 (late endosomes), and Rab11 (slow recycling compartment)[44–47]. HEK293 cells expressing β$_2$AR (Fig. 2) or V$_2$R (Fig. 3), along with Gα$_s$-GFP and various red-fluorescent Rab proteins, were analyzed using live cell confocal microscopy following agonist stimulation. Prior to stimulation, Gα$_s$ did not colocalize with any of the Rab proteins. Upon stimulation of β$_2$AR with Iso for 2 to 20 minutes, minimal or no colocalization was observed with Rab4 (Fig. 2A), and only weak and partial colocalization with Rab5 was observed overtime (Fig. 2B). In contrast, strong colocalization between Rab7 and Rab11 was detected as early as 2 min post-stimulation, with this colocalization increasing over time

(Fig. 2C, D). Quantitative analysis using Manders' overlap coefficient showed that after agonist stimulation, colocalization with Rab4 and Rab5 remained weak, whereas significantly higher and rapid colocalization was observed with Rab7 and Rab11, particularly with Rab7 at early time points (Fig. 4A). Colocalization analysis was similarly conducted in V$_2$R-expressing cells stimulated with AVP for 0 to 20 minutes (Figs. 3, 4B). Little colocalization was observed between Gα$_s$ and Rab4 while only weak colocalization was detected with Rab5 (Fig. 3A, B and Fig. 4B). Conversely, pronounced and progressively increasing colocalization with both Rab7 and Rab11 was apparent from 2 minutes post-stimulation onward (Fig. 3C, D). Quantitative analysis confirmed that the colocalization of Gα$_s$ with Rab11 was significantly higher than that observed with the Rab4 and Rab5 (Fig. 4B).

To corroborate our microscopy results, we performed enhanced bystander BRET (ebBRET) assays to quantify the proximity between the donor Renilla Luciferase II tagged to Gα$_s$ (Gα$_s$-RLucII) and the acceptor Renilla reniformis GFP anchored to various Rab proteins (rGFP-Rabs) in HEK293 cells expressing either β$_2$R or V$_2$R (Fig. 5A). Although RLucII and rGFP can self-associate with moderate (micromolar) affinity[48,49], this intrinsic interaction actually facilitates efficient BRET transfer and has been leveraged to generate sensitive trafficking sensors (Namkung et al.[48]). Therefore, although this intrinsic interaction may contribute to basal BRET signals, dynamic changes in Gα$_s$ recruitment to endosomes are nonetheless

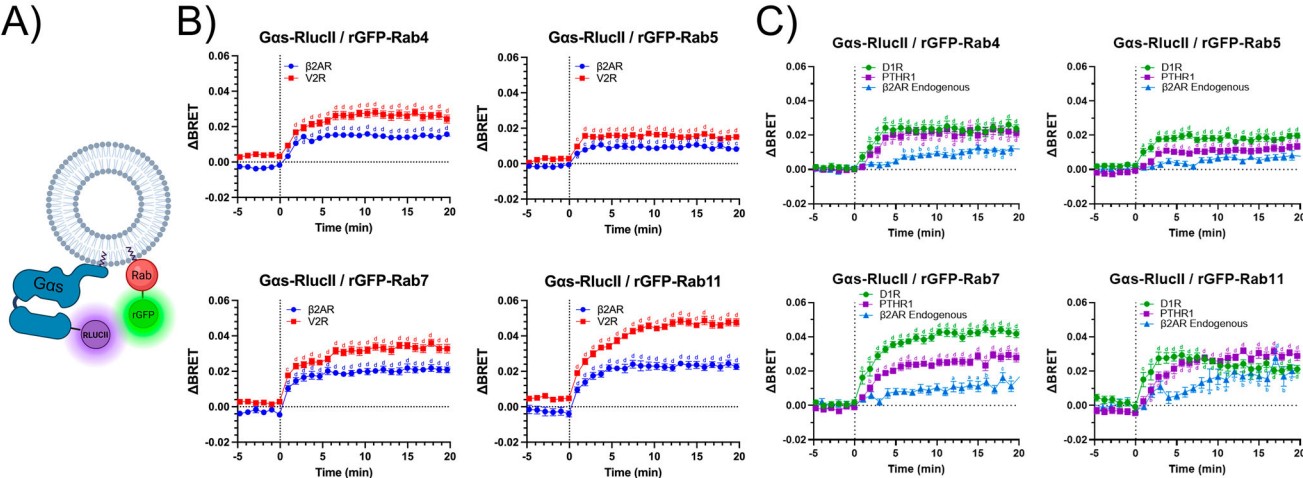

**Fig. 5 | Gα$_s$ Recruitment on the Different Endosomes Following Stimulation of β$_2$AR or V$_2$R. A** Illustration of ebBRET-based biosensor used to monitor the recruitment (proximity) of Gα$_s$-RlucII to rGFP-tagged Rab proteins bound to endosomes. Created with Biorender.com. **B,C** Time course of Gα$_s$ translocation to Rab4, Rab5, Rab7, and Rab11 labeled endosomes following **B** β$_2$AR stimulation with 1 μM Iso or V$_2$R stimulation with 1 μM AVP and **C** D$_1$R stimulation with 100 μM dopamine or PTHR1 stimulation with 100 nM PTH1-34 or endogenous β$_2$AR stimulation with 1 μM Iso, monitored using the ebBRET biosensor assay described in

(**A**). The vertical dotted line indicates the time of agonist addition. Data are presented as average of normalised BRET values (ΔBRET) of means ± SEM of triplicate measurements, pooling the data from four experiments. The BRET ratios corresponding to these data are presented in Supplementary Figs. 2 and 7. Statistical analysis was conducted using one-way ANOVA with Dunnett's multiple comparisons test where each time point was compared to 0 min. Significance levels indicated as a, $p < 0.05$; b, $p < 0.01$; c, $p < 0.001$; d, $p < 0.0001$.

expected to result in increased BRET signals. Upon agonist stimulation, we observed a rapid and significant increase in ΔBRET with all Rab proteins, with rises detectable as early as 1 minute and sustained throughout the 20-minute observation period (Fig. 5B, Supplementary Fig. 2). Gα$_s$ exhibited the strongest and most sustained ebBRET responses with Rab7 and Rab11 for both β$_2$AR and V$_2$R, indicating prominent and persistent recruitment to late and recycling endosomal compartments. In contrast, responses with Rab4 were moderate and those with Rab5 were minimal for both receptors. Because BRET efficiency can be affected by factors such as probe orientation and intracellular localization (including local protein environment, membrane curvature and surface area of the compartment) of each biosensor, the results from distinct BRET pairs targeted to different endosomal compartment cannot be directly compared. Therefore, the data are interpreted qualitatively, with emphasis on overall trafficking patterns rather than absolute quantitative differences between Rab-labeled compartments. Together with the imaging results, these findings support the rapid and robust recruitment of Gα$_s$ to rab7 and rab11-labeled endosomes upon activation of both GPCRs.

To compare these findings to the distribution of receptors within the endocytic pathway, we monitored the internalization of cell surface-labeled SNAP-tagged β$_2$AR and V$_2$R in Rab-labeled compartments (Rab4, Rab5, Rab7, and Rab11) following agonist stimulation using live cell confocal microscopy (Supplementary Figs 3–5). Colocalization analysis and quantification revealed that both β$_2$AR and V$_2$R rapidly and robustly colocalize with Rab5-positive early endosomes shortly after agonist stimulation. Over time, β$_2$AR shows progressively increased colocalization with Rab4 and Rab11 recycling endosomes but does not colocalize with Rab7-positive late endosomes. Conversely, V$_2$R exhibits minimal colocalization with Rab4 and Rab11 but shows a time-dependent increase in colocalization with Rab7 compartments. These trafficking patterns are consistent with previous reports[50–52]. Notably, although these GPCRs are primarily localized in early endosomes (Rab5), Gα$_s$ rapidly accumulated on late endosomes (Rab7) and slow recycling endosomes (Rab11) following the stimulation of β$_2$AR and V$_2$R (Figs. 2–4), highlighting a distinct endosomal distribution pattern.

Gα$_s$ associates with Gβγ subunits at the PM, and recent studies have reported that Gβγ enhances the early endosomal translocation of Gα$_s$ following V$_2$R stimulation[41]. To determine whether Gβγ influences the

redistribution of Gα$_s$ across different Rab-labeled endosomal compartments, we compared the localization of Gα$_s$-GFP in cells with or without Gβ1γ2 overexpression after V$_2$R activation, using both confocal microscopy and ebBRET assays (Supplementary Fig. 6). Our analyses indicate that Gβγ overexpression did not significantly affect the pattern or extent of Gα$_s$ redistribution among the various Rab-positive endosomes. Consequently, Gβγ overexpression was not included in subsequent analyses.

Together, these findings suggest that activation of both β$_2$AR and V$_2$R induces a rapid and sustained redistribution of Gα$_s$ to late (Rab7) and slow recycling (Rab11) endosomes—a process that is not significantly influenced by Gβγ subunits—and is distinct from the receptors' predominant localization to early (Rab5) endosomes. This indicates that receptors and Gα$_s$ follow distinct trafficking pathways.

### Gα$_s$ endosomal redistribution induced by other class A and B GPCRs and endogenous GPCR

To determine whether the endosomal redistribution of Gα$_s$ is a general feature of Gα$_s$-coupled GPCRs, we extended our analysis to the Class A dopamine D1 receptor (D$_1$R) and the Class B parathyroid hormone receptor 1 (PTHR1)[28,53]. Using the same Gα$_s$-RLucII/rGFP-Rab ebBRET assay (Fig. 5A), we assessed Gα$_s$ recruitment to Rab4-, Rab5-, Rab7-, and Rab11-positive compartments in HEK293 cells overexpressing either D$_1$R or PTHR1. Upon agonist stimulation, a rapid and significant ΔBRET increase was detected across all Rab-labeled endosomes as early as 1–3 min for both receptors (Fig. 5C, Supplementary Fig. 7). Notably, the most pronounced and sustained ebBRET responses were observed with Rab7 and Rab11, whereas responses were moderate with Rab4 and minimal with Rab5. These trafficking patterns closely resembled those previously observed with β$_2$AR and V$_2$R, underscoring a consistent and selective recruitment of Gα$_s$ to late and recycling endosomal compartments, regardless of receptor class.

In addition, we investigated whether activation of endogenous GPCRs could induce similar Gα$_s$ trafficking patterns. Given the well-established endogenous expression of β$_2$AR in HEK293 cells, we employed the Gα$_s$-RLucII/rGFP-Rab ebBRET assay to monitor Gα$_s$ dynamics following stimulation of the native receptor (Fig. 5C, Supplementary Fig. 7)[54]. Isoproterenol treatment led to a progressive and sustained increase in ΔBRET

with Rab4, Rab7, and Rab11 compartments, reaching statistical significance compared to baseline (t = 0) at 6, 12, and 8 minutes, respectively. However, no significant change was detected with Rab5. Although endosomal recruitment of $G\alpha_s$ induced by endogenous $\beta_2AR$ activation was weaker and occurred with slower kinetics than in overexpression systems, the overall response profile—characterized by moderate recruitment to Rab4 and more robust association with Rab7 and Rab11—was consistent with our earlier findings for heterologously expressed receptors. Of note, we were unable to detect $G\alpha_s$-GFP redistribution by confocal microscopy following stimulation of endogenous $\beta_2AR$, possibly due to lower signal intensity associated with native receptor activation levels. These results highlight the generalizability and physiological relevance of $G\alpha_s$ endosomal redistribution in response to GPCR activation, extending this phenomenon beyond heterologous expression models.

## Impact of GPCR activation and internalization on $G\alpha_s$ trafficking/translocation

To investigate the impact of GPCR activation and internalization on the redistribution of $G\alpha_s$, we analyzed the effects of a panel of synthetic and natural $\beta_2AR$ ligands on $G\alpha_s$-GFP translocation using live cell confocal microscopy (Fig. 6). Epinephrine (Epi) is a natural full agonist, demonstrating similar efficacy, potency, and capacity for $\beta_2AR$ internalization[55] as the synthetic full agonist Iso, which was used in Fig. 1A. The kinetics and extent of $\beta_2AR$ internalization along with the redistribution of $G\alpha_s$ to punctate structures following Epi stimulation, closely resemble those observed with Iso (Fig. 6A). Salmaterol (Sal) is classified as a high affinity synthetic partial agonist, exhibiting lower efficacy but higher potency than both Iso and Epi. However, it induces $\beta$-arrestin recruitment less effectively and drives $\beta_2AR$ internalization approximately 50% less than Iso[55,56]. Following stimulation with Sal, we observed a decrease in $\beta_2AR$ internalization levels; however, the redistribution of $G\alpha_s$ appeared unaffected (Fig. 6B). Dobutamine (Dob) is a synthetic $G\alpha_s$-biased agonist that is equally potent in activating $G\alpha_s$ as Epi, but it does not recruit $\beta$-arrestin and therefore does not promote $\beta_2AR$ internalization[56]. Notably, after stimulation with Dob, $\beta_2AR$ was not internalized, yet $G\alpha_s$ was still rapidly translocated to punctate structures (Fig. 6C).

Given the qualitative nature of the confocal imaging data, we sought to quantitatively validate $G\alpha_s$ trafficking from the PM to endosomes across multiple cells using ebBRET assays. We first analyzed the kinetic of $G\alpha_s$ dissociation from the PM in HEK293 cells expressing $\beta_2AR$ after stimulation with the various agonists over a 0–20 min timeframe. This analysis was performed by measuring the ebBRET signal between $G\alpha_s$-RlucII and rGFP anchored at the PM via a prenylated CAAX motif (rGFP-CAAX) (Fig. 6D)[41,48]. As expected, $\beta_2AR$ activation triggered a rapid and robust decrease in the BRET signal at the PM (Fig. 6E), reflecting $G\alpha_s$ dissociation from the PM, as previously reported[34,35,41]. All agonists induced similar dissociation kinetics ($t_{1/2} \approx 1$ min for all ligands), with maximal dissociation observed within 3–5 min post-stimulation. No significant differences in rate or extent of PM dissociation were found among the agonists tested. We next monitored $G\alpha_s$ translocation to late endosomes using the Rab7-based ebBRET assay (Fig. 6G). This analysis revealed a rapid and sustained accumulation of $G\alpha_s$ on Rab7-positive endosomes in response to agonist treatment (Fig. 6H), with $t_{1/2} \approx 3$ min for all ligands. Notably, $G\alpha_s$ endosomal accumulation occurred at a slower rate than PM dissociation, but no significant differences in kinetics or magnitude were observed between ligands. Similar results were also obtained for $G\alpha_s$ trafficking to other Rab-labeled endosomes (Supplementary Fig. 8), with no significant ligand-dependent differences.

To determine whether $G\alpha_s$ trafficking is dependent on agonist concentration, we next performed concentration–response experiments using both CAAX- and Rab7-based ebBRET assays (Fig. 6F, I). These assays revealed a concentration-dependent relationship for all ligands, with increasing agonist concentrations driving both a greater dissociation of $G\alpha_s$ from the PM (CAAX) and a corresponding increase in its recruitment to late endosomes (Rab7). For each ligand, the potencies (logEC50 values) were

highly similar between the two compartments, indicating that the concentration required to promote $G\alpha_s$ departure from the PM closely mirrors that required for its accumulation on late endosomes: Iso: –9.79 (CAAX) vs –9.50 (Rab7); Epi: –8.74 (CAAX) vs –8.50 (Rab7); Sal: –9.39 (CAAX) vs –9.33 (Rab7); and Dob: –7.44 (CAAX) vs –7.54 (Rab7). Notably, dobutamine displayed a lower potency overall compared to the other ligands, consistent with previous reports of its weaker ability to stimulate $\beta_2AR$-mediated $G\alpha_s$ recruitment and cAMP production[56]. Together, these results suggest that $G\alpha_s$ redistribution is concentration-dependent, with maximal trafficking occurring at higher ligand concentrations, and that endosomal accumulation of $G\alpha_s$ directly reflects the extent of receptor activation at the PM for different $\beta_2AR$ agonists.

To further explore whether $G\alpha_s$ trafficking to endosomes requires GPCR internalization, we expressed a dominant-negative mutant of dynamin (DynK44A), which inhibits both clathrin- and caveolin-mediated endocytosis[57,58]. Confocal microscopy confirmed that, following 10 min of Iso stimulation, $\beta_2AR$ internalization was completely blocked in cells expressing DynK44A. In contrast, $G\alpha_s$ redistribution to intracellular vesicular structures remained unaffected by DynK44A expression (Fig. 7A). Consistent with these observations, results from both CAAX- and Rab7-based ebBRET assays confirmed that neither $G\alpha_s$ dissociation from the plasma membrane (Fig. 7B) nor its accumulation on Rab7-positive endosomes (Fig. 7C) was affected by DynK44A expression. In contrast, endocytosis of $\beta_2AR$-RlucII into rGFP–FYVE-labeled early endosome was strongly inhibited under these conditions (Fig. 7D). Together, these results demonstrate that $G\alpha_s$ trafficking to endosomal compartments occurs independently of $\beta_2AR$ internalization via classical dynamin-dependent endocytic pathways.

These results indicate that $G\alpha_s$ rapidly translocate to endosomes through a mechanism independent of GPCR internalization, and it is the degree of receptor activation—not the extent of internalization—that govern $G\alpha_s$ dissociation from the PM and its subsequent trafficking to endosomes.

## Influence of Plasma Membrane Association on $G\alpha_s$ Trafficking

The rapid dissociation of $G\alpha_s$ from the PM and its subsequent accumulation on late endosomes is consistent with previous reports that receptor activation induces $G\alpha_s$ depalmitoylation, enabling its release from the PM into the cytoplasm and subsequent trafficking to intracellular compartments[34,35]. To investigate whether persistent PM association or complete cytosolic localization affects $G\alpha_s$ translocation to endosomes upon receptor stimulation, we used two variants: a myristoylated $G\alpha_s$ (Myr-$G\alpha_s$), which remains constitutively anchored to the PM, and a cytosolic form with a mutated palmitoylation site, $G\alpha_s$(C3S)[34,38]. We first assessed the functional activity of these $G\alpha_s$ constructs using a BRET-based cAMP assay with the EPAC biosensor in $G\alpha_s$ knockout HEK293 cells (HEK $G\alpha_s$-KO) expressing $\beta_2AR$ (Supplementary Fig. 9). As expected, the absence of $G\alpha_s$ abolished $\beta_2AR$-stimulated cAMP production. In cells expressing Myr-$G\alpha_s$, isoproterenol stimulation increased cAMP levels, confirming that this PM-restricted variant remains functionally competent. Additionally, these cells also displayed a lower baseline BRET signal compared to wild-type $G\alpha_s$, suggestive of elevated basal cAMP and possible constitutive activity, as previously reported[59]. Notably, stimulation of cells expressing $G\alpha_s$(C3S) led to a partial increase in cAMP compared to wild-type $G\alpha_s$, indicating that even the non-palmitoylated, cytosolic form retains the capacity to be activated by $\beta_2AR$. We next examined the subcellular distribution of these constructs, using GFP-tagged versions visualized by live-cell confocal microscopy in $\beta_2AR$-expressing HEK293 cells (Fig. 8A). Upon agonist stimulation, Myr-$G\alpha_s$-GFP remained restricted to the PM and did not form cytoplasmic vesicles, while $G\alpha_s$(C3S)-GFP displayed a diffuse cytoplasmic distribution both before and after stimulation, with no detectable vesicular relocalization. These observations were further supported by ebBRET assays, which confirmed that Myr-$G\alpha_s$-GFP did not significantly dissociate from the PM (Fig. 8B) and that $G\alpha_s$(C3S) did not accumulate appreciably on PM (Fig. 8B) and Rab7 endosomes (Fig. 8C).

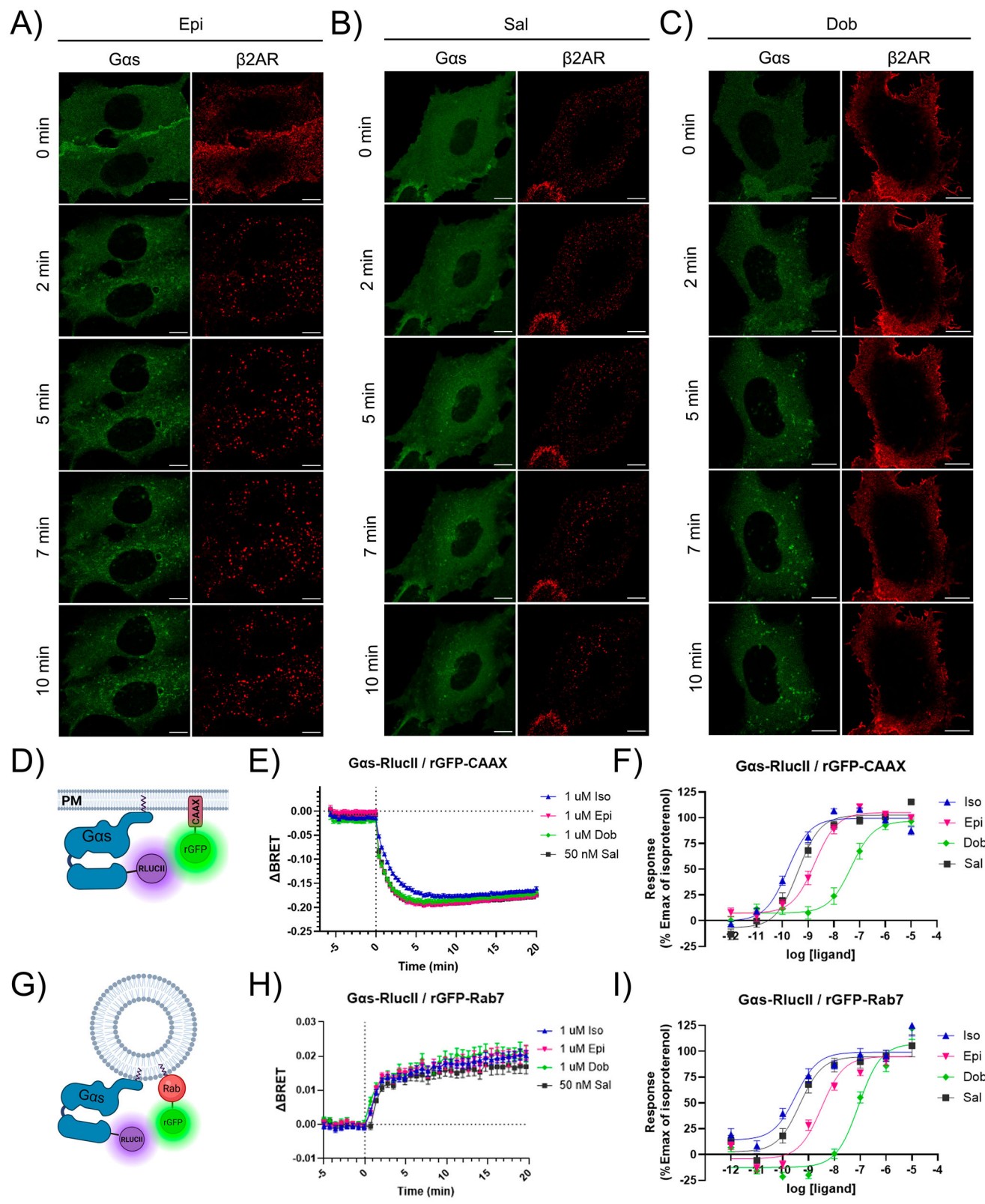

Taken together, these findings indicate that release from the PM is a prerequisite for Gα$_s$ trafficking to endosomes, and that simple activation of cytosolic Gα$_s$(C3S) is not sufficient to drive its endosomal localization. This suggests a critical role for palmitoylation-driven membrane association in enabling Gα$_s$ redistribution to endosomal compartments upon receptor activation.

## Discussion

The fate and trafficking of GPCRs have been extensively studied; however, little is currently known about the trafficking pathways of the heterotrimeric G proteins, which dissociate from their receptors upon activation. This knowledge is essential, as it is now well established that GPCRs can signal through these G proteins from various intracellular compartments, notably

**Fig. 6 | Effects of $\beta_2$AR Stimulation by different Agonists on G$\alpha_s$ Internalization and Trafficking. A–C** Serum starved HEK293 cells co-expressing Flag-$\beta_2$AR and G$\alpha_s$-EGFP were incubated at 37 °C for 15 min. with 1:500 anti-Flag and 1:1000 AF594. Cells were next stimulated with the indicated agonists and imaged by live-cell confocal microscopy: **A** 10 µM epinephrine (Epi), **B** 50 nM salmeterol (Sal), and **C** 10 µM dobutamine (Dob). Scale bars, 10 µm. **D** Illustration of the ebBRET-based biosensor assay for monitoring proximity between G$\alpha_s$-RlucII and plasma membrane-anchored rGFP-CAAX (created with BioRender.com). **E** Time course of G$\alpha_s$ dissociation from the PM following $\beta_2$AR activation by various agonists, as measured by the assay described in (**D**). The vertical dotted line indicates agonist addition. Data represent normalized BRET values ($\Delta$BRET), shown as the mean ±

SEM of triplicates; N = 3. **F** Concentration–response curves depicting G$\alpha_s$ dissociation from the PM in response to $\beta_2$AR stimulation by the indicated agonists and measured by the assay described in (**D**). Data are mean ± SEM; N = 3. **G** Illustration of of the ebBRET biosensor assay for detecting recruitment (proximity) of G$\alpha_s$-RlucII to rGFP-labeled Rab7 endosomes (created with BioRender.com). **H** Time course showing G$\alpha_s$ translocation to Rab7-labeled endosomes after $\beta_2$AR stimulation with different agonists, monitored using the biosensor in (**G**). The vertical dotted line marks agonist addition. Data show normalized BRET values ($\Delta$BRET), as mean ± SEM; N = 4. **I** Concentration–response curves for G$\alpha_s$ recruitment to Rab7-positive endosomes upon $\beta_2$AR stimulation with the indicated agonists, and measured by the assay described in (**G**). Data are mean ± SEM; N = 4.

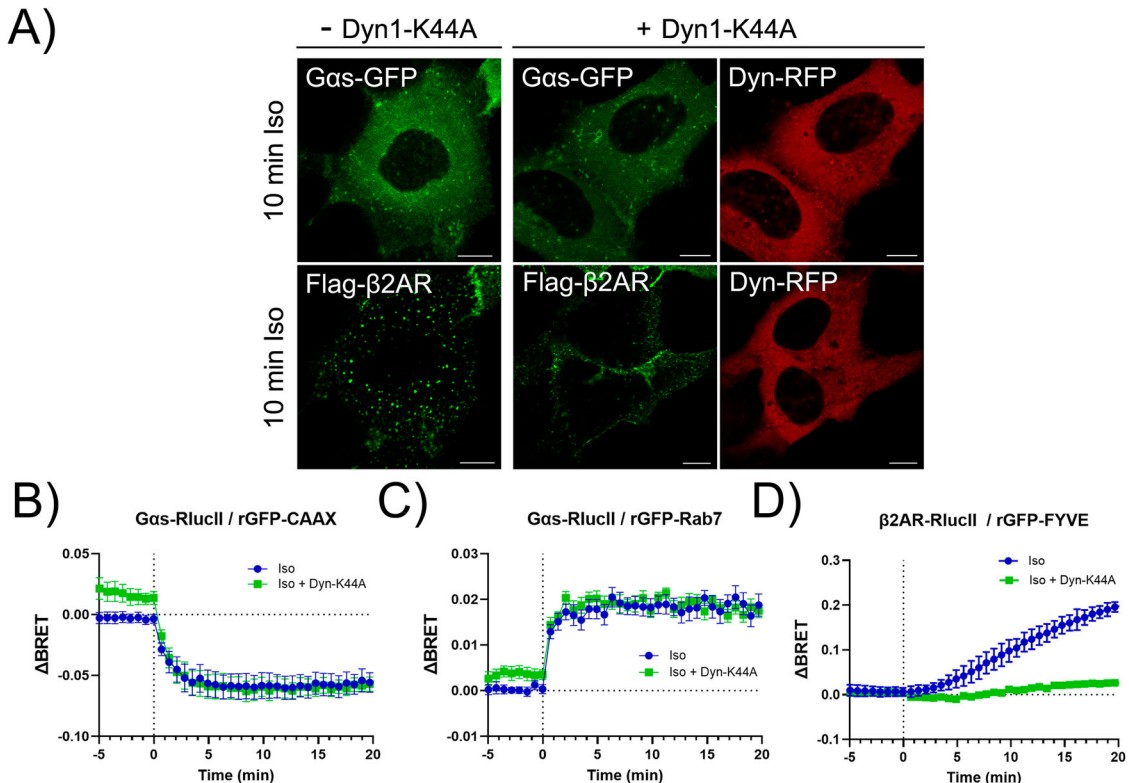

**Fig. 7 | G$\alpha_s$ Internalization Occurs Independently of Receptor Internalization. A** Live-cell confocal microscopy images of HEK293 cells co-expressing G$\alpha_s$-EGFP and Flag-$\beta_2$AR, with or without co-expression of Dyn1(K44A)-mRFP. Cells were serum-starved for 1 h at 37 °C and then stimulated with 10 µM isoproterenol (Iso) for 10 min at 37 °C. As a control, Flag-$\beta_2$AR were labeled with 1:500 anti-Flag and 1:1000 AF488 at 37 °C for 15 min prior to stimulation. Scale bars, 10 µm. **B** Time course of G$\alpha_s$ dissociation from the PM following $\beta_2$AR activation by 100 nM Iso, monitored using the ebBRET biosensor assay described in Fig. 6D, in the absence or presence of Dyn1(K44A) overexpression. The vertical dotted line indicates addition

of the agonist. N = 3. **C** Time course of G$\alpha_s$ translocation to Rab7-endosomes following $\beta_2$AR stimulation with 100 nM Iso, measured with the ebBRET biosensor described in Fig. 6G, in the absence or presence of Dyn1(K44A) overexpression. N = 3. **D** ebBRET experiment was used to measure time course of $\beta_2$AR translocation to FYVE-endosomes following $\beta_2$AR stimulation with 100 nM Iso, in the absence or presence of Dyn1(K44A) overexpression. N = 3. All ebBRET data are presented as normalized BRET values ($\Delta$BRET), shown as mean ± SEM of triplicate measurements.

early endosomes, resulting in distinct physiological outcomes. In this study, we combined live-cell confocal microscopy and ebBRET assays to characterize the spatiotemporal organization of G$\alpha_s$ within specific compartments of the endosomal pathway following GPCR activation, with the goal of uncovering mechanistic insights into its endosomal recruitment. Our results reveal that, across both class A ($\beta_2$AR, D$_1$R) and class B (V$_2$R, PTHR1) GPCRs, G$\alpha_s$ is rapidly and preferentially translocated to Rab7-labeled late endosomes and Rab11-labeled slow-recycling endosomes within minutes of stimulation. In contrast, G$\alpha_s$ shows low colocalization with early endosomal markers and only partial and transient colocalization with the internalized receptors. Furthermore, G$\alpha_s$ activation at the PM, dissociation from the PM, and its palmitoylation site are essential for

subsequent endosomal translocation, which occurs independently of receptor internalization.

Numerous G$\alpha_s$-coupled GPCRs show sustained signaling by facilitating a second round of agonist-dependent allosteric coupling to G$\alpha_s$ at the endosomal membrane. This endosomal signaling is particularly pronounced for class B receptors, such as V$_2$R, which have been reported to form a stable megacomplex containing GPCR–$\beta$-arrestin–G$\alpha_s$ within early endosomes[23,28,60]. To investigate whether the propensity of a GPCR to signal from endosomes influences G$\alpha_s$ translocation/internalization, we compared the kinetics and pathways of G$\alpha_s$ internalization following stimulation of prototypical class A ($\beta_2$AR) and class B (V$_2$R) GPCRs using time-frame live cell confocal microscopy and ebBRET. Our findings revealed no significant differences in the kinetics or levels of G$\alpha_s$ translocation to vesicles/

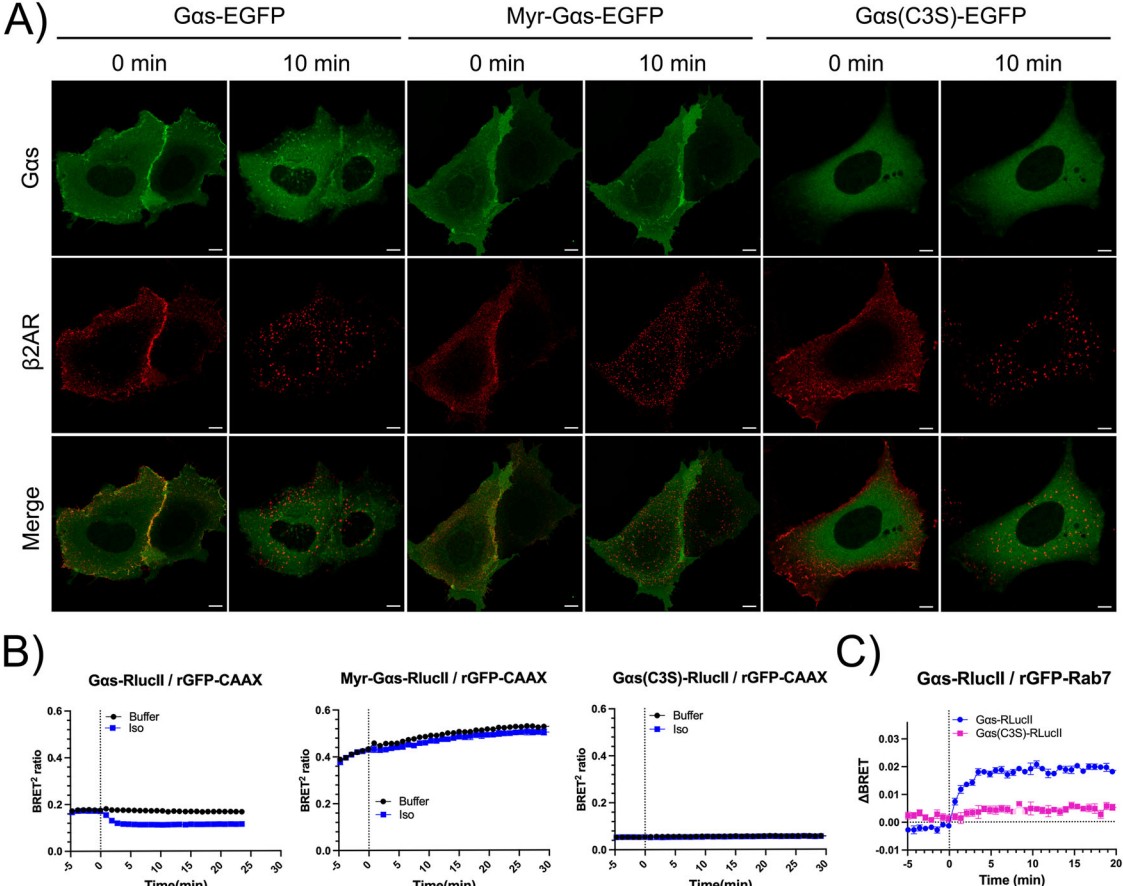

**Fig. 8 | Impact of Myristoylation and palmitoylation on Gαs Trafficking. A** Live-cell confocal images of serum-starved HEK293 cells co-expressing Flag-β2AR and either wild-type Gαs-EGFP, myristoylated Gαs-EGFP (Myr-Gαs-EGFP), or palmitoylation-deficient Gαs(C3S)-EGFP. Cells were labeled at 37 °C with anti-Flag antibody (1:500) and AF594 (1:1000) for 15 min, followed by stimulation with 10 μM Iso for 10 min. Scale bars, 10 μm. **B** Time course of PM dissociation for wild-type Gαs, Myr-Gαs, and Gαs(C3S) following β2AR activation with 1 μM Iso, monitored using the ebBRET biosensor assay described in Fig. 6D. The dotted line indicates agonist addition. Data represent the mean ± SEM of triplicates; N = 3. **C** Time course of wild-type Gαs and Gαs(C3S) translocation to Rab7-endosomes following β2AR stimulation, measured with the ebBRET biosensor described in Fig. 6G. The vertical dotted line marks agonist addition. Data are shown as normalized BRET values (ΔBRET), mean ± SEM of triplicates; N = 3.

endosomes between the two GPCRs. In both instances, Gαs displayed rapid kinetics of dissociation from the PM (~1 min), with subsequent accumulation on vesicles/endosomes occurring approximately 2–4 minutes post-stimulation and increasing over time. These results are consistent with previous studies showing that Gαs dissociates from the PM or translocates to vesicles within 1–5 min following β2AR stimulation[33,34,39,61].

Gαs has been previously localized on early endosomes by both microscopy and BRET-based approches[11,21,23,41,61]. However, the precise kinetics of its redistribution across distinct endocytic compartments following activation of various GPCRs had not been systematically characterized prior to this study. Our confocal microscopy analyses revealed that, while Gαs is present on Rab5-labeled early endosomes, this colocalization is relatively weak and only partial. In contrast, we observed robust and rapid colocalization of Gαs with Rab7-labeled late endosomes and Rab11-labeled slow-recycling endosomes as early as 2 minutes post-stimulation. These findings were confirmed by ebBRET assays, which showed a pronounced and rapid (1–3 min) recruitment of Gαs to Rab7 and Rab11 compartments, but only modest association with Rab5 and Rab4 early endosome, following stimulation of both class A (β2AR, D1R) and class B (V2R, PTHR1) GPCRs. Notably, stimulation of endogenous β2AR yielded qualitatively similar Gαs endosomal trafficking, though with reduced magnitude and slower kinetics, underscoring the generalizability and physiological relevance of this process. This finding suggests that upon

activation and dissociation from the PM, Gαs traffics directly to late endosomes and slow-recycling endosomes, bypassing the classical endocytic route, and instead translocating directly from the cytoplasm. Consistent with our findings, the group of Berlot previously reported, using fluorescence microscopy, that Gαs did not colocalize with the early endosomal marker RhoB or with β2AR following stimulation; however, it exhibited high abundance on Rab11 recycling endosomes[39]. Of note, their study did not examine other endocytic markers. Furthermore, a recent study has demonstrated that the endogenous Gβγ dimer is predominantly localized on rab7-positive late endosomes[61], suggesting that Gα subunits may also be present in these compartments. As proposed by these research groups, these localization on rab7- and rab11- labeled endosomes could play a role in the degradation or recycling of G proteins. Our group has previously identified a role for Gαs in endosomal sorting of receptors toward the degradation pathway through interaction with the ESCRT machinery[62,63], raising the possibility that Gαs recruitment on late endosomes may be involved in this mechanism.

Because it is well established that GPCRs can stimulate Gαs-dependent signaling from early endosomal compartments, we expected a high degree of colocalization between the internalized receptors and Gαs following stimulation, particularly for class B receptors such as V2R. However, our observations revealed only minimal colocalization between Gαs and both β2AR and V2R at all examined time points, as well as different trafficking

kinetics across endosomal compartments, indicating that they do not traffic together. Live cell imaging suggests a partial colocalization within the vesicles, indicating a potential side-by-side arrangement on endosomal microdomains and/or rapid transient colocalization. This finding aligns with previous reports indicating that $G\alpha_s$ transiently enters and exits clathrin- and caveolin-coated vesicles without accumulating within them[64]. Furthermore, the Nb37 nanobody, which detects the active form of $G\alpha_s$, has been found to localize within specific microdomains of endosomes, such as actin/sorting nexin/retromer tubular (ASRT) microdomains, with only partial colocalization with the $\beta_2AR$[11,65].

Several mechanisms could reconcile our findings with the abundant reports of early endosomal $G\alpha_s$ signaling. One possibility is that transient or partial encounters between $G\alpha_s$ and GPCRs within early endosomes are sufficient to trigger effective signaling, even if colocalization appears minimal by microscopy or BRET assays. Another explanation, supported by recent studies, is that a pool of $G\alpha_s$ already exists on endosomal membranes prior to receptor activation[61]. In this scenario, internalized GPCRs could activate this pre-existing $G\alpha_s$ population without a need for substantial new trafficking of $G\alpha_s$ from the PM. Indeed, our observation of strong localization of $\beta_2AR$ and $V_2R$ to early endosomes supports the anatomical prerequisites for this model. It is also possible that $G\alpha_s$ is activated at the PM and then translocate to endosomes, carrying its activated state but acting independently of direct engagement with internalized GPCRs. This mechanism has been suggested for $G\alpha_i$ and could explain the presence of active $G\alpha_s$ within endosomes that lack their cognate receptors[66]. Importantly, our results also raise the possibility that $G\alpha_s$-dependent signaling can originate from late and slow-recycling endosomes themselves, not just early endosomes. Although these compartments are not typically recognized as significant signaling hubs like early endosomes, some signaling activity or proteins have been observed within these compartments[67,68]. This suggests a broader spatial landscape for compartmentalized $G\alpha_s$ signaling, potentially contributing to distinct downstream outcomes depending on the endosomal context. Further studies will next be done to directly assess the activation state of $G\alpha_s$ and downstream signaling outputs within these compartments.

Our results demonstrate that the translocation of $G\alpha_s$ from the PM to endosomal compartments is strictly dependent on its activation at the PM and directly correlates with the level of GPCR activation, yet it occurs independently of receptor internalization. Stimulation of $\beta_2AR$ with a variety of agonists—including the endogenous full agonist epinephrine, the synthetic full agonist isoproterenol, the partial agonist salmeterol, and the $G\alpha_s$-biased agonist dobutamine—resulted in comparable kinetics and magnitudes of $G\alpha_s$ dissociation from the plasma membrane and accumulation on endosomes, with no significant differences among these ligands. Additionally, neither expression of the dominant-negative dynamin mutant Dyn(K44A), which blocks classical receptor endocytosis, nor stimulation with dobutamine, which fails to recruit $\beta$-arrestin and does not promote $\beta_2AR$ internalization[39,41,69], altered $G\alpha_s$ trafficking. These findings highlight that $G\alpha_s$ translocation operates through a mechanism distinct from that of GPCR internalization. While we did not directly ablate $\beta$-arrestin function, our findings with dobutamine strongly indicate that $\beta$-arrestin engagement is not required for $G\alpha_s$ endosomal translocation, at least in the context of $\beta_2AR$. This is consistent with recent studies reporting that $\beta$-arrestins are dispensable for $G\alpha_s$ translocation to early endosomes[12,59,70]. Furthermore, although recent studies have suggested that, for receptors like $V_2R$, $\beta$-arrestin and $G\beta\gamma$ complexes can facilitate $G\alpha_s$ trafficking to early endosomes[41], our data show no appreciable difference in $G\alpha_s$ endosomal translocation between class A and class B GPCRs, nor upon $G\beta1\gamma2$ overexpression. These discrepancies with previous reports may reflect methodological differences (e.g., we did not employ $\beta$-arrestin supression or $G\beta\gamma$ scavenger tools), as well as possible differences in BRET biosensors used.

As shown for $G\alpha_s$ and $G\alpha_i$, activation of $G\alpha_q$ by GPCRs at the PM also induces its translocation to endosomes, independent of $\beta$-arrestin engagement and receptor endocytosis. In contrast, the activity of $G\alpha_q$ within endosomes has been found to depend on both receptor endocytosis-dependent and -independent mechanisms[71]. This suggests that part of $G\alpha_q$ trafficking may involve cytoplasmic release followed by targeting to endomembranes—a mechanism that parallels the proposed pathway for $G\alpha_s$ translocation. Both $G\alpha_q$ and $G\alpha_i$ have been reported to colocalize with early endosomal markers (Rab5 and FYVE) by microscopy and BRET assays[66,71]. It would be interesting to investigate whether $G\alpha_q$ and $G\alpha_i$ can also be found on Rab7- and Rab11- labeled endosomes.

Recent findings suggest that the $G\alpha_i$ subunit, known to be both palmitoylated and myristoylated, can rapidly localize to endosomes upon activation[66]. This prompted us to investigate whether the PM dissociation is essential for $G\alpha_s$ translocation to late endosomes. To address this, we compared the behavior of myristoylated $G\alpha_s$ (Myr-$G\alpha_s$), which is constitutively anchored to the PM, and a non-palmitoylated cytosolic variant, $G\alpha_s$(C3S). Upon $\beta_2AR$ stimulation, neither construct efficiently translocated to endosomes; however, both remained functionally competent in signaling, as evidenced by their ability to be activated by GPCRs and stimulate adenylate cyclase to generate cAMP. Consistent with previous reports[38], we observed that Myr-$G\alpha_s$ can produce elevated basal cAMP levels, likely reflecting constitutive activity due to persistent association with the PM and possible conformational effects. Activation of $G\alpha_s$(C3S), in contrast, is likely the result of stochastic encounters with active GPCRs in the cytosol. Therefore, our results suggest that activation at the PM, followed by release into the cytoplasm, is necessary for efficient $G\alpha_s$ trafficking to endosomal compartments. Our data further indicate that cytoplasmic activation alone does not promote endosomal accumulation of $G\alpha_s$, suggesting the potential requirement for dynamic palmitoylation cycles in this process. In this context, we propose that while $G\beta\gamma$ and $\beta$-arrestin may be involved in $G\alpha_s$ endosomal recruitment, $G\alpha_s$ likely requires repalmitoylation to effectively bind to intracellular membranes or to return to the PM, as previously suggested[35,38,39]. This raises the question regarding whether palmitoylation of $G\alpha_s$ occurs on endosomal membranes, the identity of the palmitoyl acyltransferases, and how this process is regulated. Elucidating these mechanisms will be the focus of future investigations.

In summary, our study reveals that both class A and class B GPCRs trigger rapid and selective redistribution of $G\alpha_s$ to late (Rab7-positive) and slow-recycling (Rab11-positive) endosomes via a pathway that requires $G\alpha_s$ activation and depalmitoylation at the PM, yet operates independently of receptor internalization and $\beta$-arrestin engagement. Notably, $G\alpha_s$ exhibits only minimal colocalization with internalized receptors, underscoring distinct and asynchronous trafficking routes. These findings challenge the prevailing model of GPCR-driven G protein signaling from early endosomes, instead suggesting a more complex and highly compartmentalized signaling landscape. These insights lay a foundation for future studies to dissect the regulatory principles and physiological implications of compartment-specific G protein signaling within cells.

## Methods
### DNA Constructs
pCMV-15F11-HA-mEGFP (Frankenbody) (#129590), pmRFP-C3-Rab5 (#14437), pmRFP-C3-Rab7 (#55125), mCherry-Rab11a-7 (#55124), and pEGFP-N1-Dyn1(K44A)-mRFP (#55795) constructs were purchased from Addgene. pCDNA3.1-3xHA-$D_1R$ and pCDNA3.1-HA-$G\gamma2$ were purchased from UMR cDNA Resource Center (University of Missouri-Rolla, Rolla, USA). pRK5-myc-$V_2R$, pcDNA3.1-G-Beta-1, pcDNA3.1-PTHR1, pSNAPf-HA-$V_2R$, pcDNA3.1-$G\alpha_s$-long-WT-RlucII(67), pcDNA3.1-$G\alpha_s$-short-WT-RlucII(119), pcDNA3.1-CAAX-rGFP, pcDNA3.1-Tdr-rGFP-Rab4, pcDNA3.1-Tdr-rGFP-Rab5, pcDNA3.1-Tdr-rGFP-Rab7, pcDNA3.1-Tdr-rGFP-Rab11 and pcDNA3.1-GFP10-EPAC-RlucII, were kindly provided by Dr. Michel Bouvier (Université de Montréal, Montreal, Canada). pcDNA3-Flag-$\beta_2AR$ was a gift by Dr. Stéphane Laporte (McGill University, Montreal, Canada) and pmRFP-C3-Rab7 was a gift by Dr. Stéphane Lefrançois (Université de Montréal, Montreal, Canada). pSNAPf-$\beta_2AR$ was kindly provided by Dr. Roshanak Irannejad (University of California, San Francisco, USA). pcDNA3.1-$G\alpha_s$(C3S)-EGFP and pcDNA3.1-$G\alpha_s$(C3S) were generated by replacing cysteine with serine at position 3 of $G\alpha_s$ in the construct pcDNA3.1-$G\alpha_s$-EGFP and pcDNA3.1-

Gα$_s$-short, respectively, using the QuikChange Lightning Site-Directed Mutagenesis Kit (Agilent Technologies). pcDNA3.1-Gα$_s$-EGFP and pcDNA3.1-Gα$_s$-mini HA were generated with the primers described in Supplementary Table 1 using the Gibson Assembly cloning kit (New England Biolabs). pcDNA3.1-Gα$_s$-EGFP was based on Hynes et al. with the exception that our construct is human instead of rat[39]. pcDNA3.1-Myr-Gα$_s$-EGFP and pcDNA3.1-Myr-Gα$_s$-long-RLucII(67) were generated by adding a threonine at position 4 and a serine at position 6 of Gα$_s$ in the construct pcDNA3.1-Gα$_s$-EGFP and pcDNA3.1-Gα$_s$-long-RlucII(67), respectively, using the QuikChange Lightning Site-Directed Mutagenesis Kit (Agilent Technologies).

### Reagents and Antibody
Isoproterenol (#I6504-100MG), Epinephrine (#E4250), Salmeterol (#94749-08-3), Dobutamine (#D0676), Dopamine (#8502), Parathyroid hormone fragment human 1-34 (PTH1-34) (#P3796) and Anti-Flag rabbit (# F7425) were purchased from Sigma Aldrich (St. Louis, Missouri, USA). [Arg8]-Vasopressin (#24289(AN)) was purchased from Anaspec Inc. (Fremont, CA, US). FuGENE6 was purchased from Promega Corporation (Madison, Wisconsin, USA). Polyethylenimine (PEI) was from Polysciences (Warrington, PA, USA). Salmon sperm DNA (SSD) was from Invitrogen. Anti-Myc mouse (#2276) and anti-EEA1 rabbit 1:1000 (#3288) were purchased from Cell Signaling (Denver, Massachusetts, USA). HRP anti-rabbit IgG was purchased from Bio-Rad (Hercules, CA, USA). Anti-HA rabbit (#PRB-101P) was purchased from Covance (Princeton, New Jersey, USA). Alexa Fluor 594 against mouse (#A21203), Alexa Fluor 594 against rabbit (#A21207), Alexa Fluor 488 against mouse (#A21202) Alexa Fluor 488 against rabbit (#A21206) and salmon sperm DNA (SSD) were purchased from Invitrogen (Clarsbad, CA, USA). Horseradish peroxidase-conjugated (HRP) anti-Gα$_s$ 1:1000 (#sc-135914-HRP) were purchased from Santa Cruz (Dallas, Texas, USA) Coelenterazine 400 A was from GoldBio (St-Louis Missouri, USA), Prolume Purple was from Nanolight technology (Pinetop, Arizona, USA). SNAP-Surface 649 (#S9159S) was purchased from NEB (Ipswich, Massachusetts, USA).

### Cell Culture
HEK293SL cells were kindly provided by Dr. Stéphane Laporte (McGill University, Montreal, Quebec, Canada). HEK293T-Gα$_s$ KO (HEK293T Gα$_s$ crispr knock-out) cell lines were kindly provided by Dr Asuka Inoue (Tohoku University, Sendai, Miyagi, Japan)[72]. They were grown in high-glucose Dulbecco's modified Eagle's medium (DMEM) (Gibco, Waltham, Massachusetts, USA). The culture media was supplemented with 10% fetal bovin serum (FBS) (HyClone Laboratories, Logan, UT, USA) and 1% penicillin-streptomycin-glutamine solution (PSG) (Invitrogen, Clarsbad, CA, USA). Unless otherwise mentioned, all incubation sequences were performed at 37 °C in the presence of 5% CO2. Cells were transfected with FuGENE6 transfection reagents according to the manufacturer's instructions, or with PEI, as described in the BRET section.

### Confocal Microscopy
For live cell confocal microscopy analysis, HEK293SL cells ($3.5 \times 10^4$) were plated in precoated (poly-L-lysine solution) 35 mm glass-bottom dishes (MaTek Corporation). The day after, cells were transfected with different DNA constructs (0.5 or 0.25 μg Gα$_s$-EGFP, 0.5 μg Flag-β$_2$AR, 0.01 μg Myc-V$_2$R, 0.5 ug SNAP-β$_2$AR, 0.25 ug SNAP-V$_2$R, 0.25 ug Tdr-rGFP-Rab4, 0.5 ug Tdr-rGFP-Rab5, 0.5 ug Tdr-rGFP-Rab7, 0.25 ug Tdr-rGFP-Rab11, 0.5 μg Dyn1(K44A)-mRFP, 0.5 μg Myr-Gα$_s$-EGFP, 0.5 μg Gα$_s$(C3S)-EGFP, 0.25 μg mCherry-Rab4, 0.25 μg RFP-Rab5, 0.25 μg RFP-Rab7, 0.5 μg mCherry-Rab11, 0.5 μg Gα$_s$-mini HA, 0.5 μg Frankenbody-EGFP, 0.25 μg Gβ1 and 0.25 μg Gγ2 using FuGENE6 transfection reagent for 48 h. The day of the experiment cells were starved in DMEM without FBS and PSG for 1 h at 37 °C and then incubated with the appropriate primary and secondary antibodies for 15 min or with SNAP-649 for 20 min at 37 °C. Cells were then wash with PBS 1X and 1 ml of FluoroBrite DMEM (Gibco Waltham, Massachusetts, USA) was added to the cells before acquisition. After which,

the corresponding agonist was added directly to the cells during acquisition. To keep the cells at 37 °C with 5% CO2, petri dishes were placed in a microchamber (Okolab, NA, Italy) attached to the stage of the microscope. Cells were selected for imaging based on the subcellular distribution of Gα$_s$; only those exhibiting appropriate plasma membrane localization and expression profiles were included in the analysis, due to observed variability in Gα$_s$ cytoplasmic distribution across the cell population. Selected cells expressed all labeled proteins, had a clearly defined plasma membrane, and displayed a cytoplasmic region largely free of vesicles before the addition of agonist. Exposure times were individually optimized for each cell and detection channel. Images were acquainted with a scanning confocal microscope (Leica TCS SP8 STED DMI8, Leica Microsystems, Toronto, On, Canada) equipped with 63X/1.4 oil-immersion objective and a tunable white light laser (470 to 670 nm). The fluorophore or fluorescent protein, green fluorescent protein (GFP), red fluorescent protein (RFP), mCherry, Alexa Fluor 488, Alexa Fluor 594 and SNAP-649 were excited with the 488, 555, 587, 488, 594 and 649 nm laser lines of the white laser, respectively and emissions were detected with HyD detector. LAS AF Lite software (Leica) was used for image acquisition and analysis. The images were further processed using Adobe Photoshop v 26.8.1 (Adobe Systems, San Jose, Ca, USA). Huygens professional 23.10 was used for colocalization quantification using Mander's overlap coefficient.

### BRET measurements
HEK293SL cells were seeded in 6-well plates ($3.5 \times 105$ cells/well) and transfected with 1 μg of total DNA diluted in 100 μL Opti-MEM (adjusted with SSD) using a 3:1 ratio of linear PEI (1 mg/mL) per μg DNA. For all BRET experiments, well were transfected with either 100 ng of Flag-β$_2$AR, 25 ng of myc-V$_2$R, 150 ng PTHR1 or 3xHA-D$_1$R, as indicated. For the EPAC assays, cells were transfected with 100 ng of GFP10-EPAC-RlucII, and 100 ng of Gα$_s$ construct, as indicated. For ebBRET monitoring Gα$_s$ translocation, 100 ng of Gα$_s$-long-WT-RlucII(67) or myr-Gα$_s$-long-WT-RlucII(67) and 1000 ng of rGFP-CAAX. For the Rab assays, cells were transfected with 200 ng of Gα$_s$-short-WT-RlucII(119) and 1000 ng of either Tdr-rGFP-Rab4, Tdf-rGFP-Rab5, Tdr-rGFP-Rab7 or Tdr-rGFP-Rab11. For overexpression of Dyn(K44A), Dyn1(K44A)-mRFP (100 ng) was used. For overexpression of Gβ1 and Gγ2, each construct was transfected at 100 ng. The day following transfection, cells were seeded ($3,5 \times 10^4$ cells/well) in white opaque 96-well microplates (Perkin Elmer) and BRET experiments were performed the next day. The day of the experiment, medium was removed and replaced by the BRET buffer (10 mM HEPES, 1 mM CaCl2, 0.5 mM MgCl2, 4.2 mM KCl, 146 mM NaCl, 5.5 mM glucose, pH 7.4) and cells were maintained at 37 °C for BRET measurements. Coelenterazine 400a (final concentrations of 5 μM) or Prolume Purple (final concentrations of 1 μM) was added 5 min before BRET measurement, as indicated. For kinetic measurements, basal BRET was measured during 300 s before being stimulated as indicated in the figure legends and BRET signal was recorded each 30 s during at least 1200 s. Plates were read on the Berthold TriStar2 LB 942 Multimode Reader (Berthold, Bad Wildbad, Germany) with the energy donor filter (410 nm) for RlucII and energy acceptor filter (515 nm) rGFP-CAAX. The BRET signal (BRET²) was determined by calculating the ratio of the light intensity emitted by the acceptor over the light intensity emitted by the donor. ΔBRET was calculated by subtracting the unstimulated BRET signal from the ligand stimulated BRET signal.

### Immunoblotting
HEK293 cells were plated il 35 mm culture dishes and transfected with Gα$_s$-WT, Gα$_s$(C33) and Myr-Gα$_s$ for 48 h. Cells were washed twice with PBS and lysed in 50 mM Tris buffer pH 7.4 (Sigma-Aldrich, Saint-Louis, Missouri, USA) containing 150 mM NaCl (ThermoFisher Scientific, Waltham, Massachusetts, USA), 1% NP40 (Sigma-Aldrich, Saint-Louis, Missouri, USA), 5 mM EDTA (Sigma-Aldrich, Saint-Louis, Missouri, USA) and complete protease inhibitors (Roche, Bâle, Switzerland) for 1 h at 4 °C and then centrifuged at 15,800x$g$ for 20 min. Protein concentration of the

supernatants was dosed using Bradford assays (NEB) and samples (30 μg) were boiled in Laemmli sample buffer. The protein samples were separated on 10% SDS-polyacrylamide gel electrophoresis (PAGE) and transferred to 0.45 μm-diameter pore-size nitrocellulose membranes (PerkinElmer Life Sciences). The membranes were blocked in Tris-buffered saline (20 mM Tris-HCl (pH 7.4) and 150 Mm NaCl) containing 0.1% Tween 20 and 5% nonfat dry milk and incubated at room temperature for overnight with antibodies (HRP conjugated anti-$G\alpha_s$ 1:1000 (#sc-135914-HRP, Santa Cruz, Dallas, USA) and enhanced chemiluminescence detection reagent (Pierce). The antibodies used for loading control were anti-EEA1 rabbit 1:1000 (#3288, Cell signaling, Danvers, USA) and HRP anti-rabbit IgG 1:7000 (Bio-Rad, Hercules, USA).

## Statistics and Reproducibility

For BRET experiments, data were analyzed in GraphPad Prism 10.5 software and data are represented as the means ± SEM of triplicate measurements in a representative experiment that was repeated at least three times with similar results. Each live-cell confocal microscopy experiences were repeated at least four times. Statistical analyses were performed using One-way ANOVA with the Tukey's multiple comparison test (for Fig. 4 and Supplementary Fig. 5) or with Dunnett's multiple comparisons test (for Fig. 5). Welch's t test was also used for Supplementary Fig. S6. Data were considered significant when $p$ values were < 0.05.

## Reporting summary

Further information on research design is available in the Nature Portfolio Reporting Summary linked to this article.

## Data availability

All relevant data supporting the findings are available within the paper and the Supplementary Materials. Source data used for generating the plots in the main figures are available in the Supplementary Data file associated with the manuscript. Additional information and reagents are available from the corresponding author upon reasonable request.

## Code availability

No custom computer code was used within this study and all commercially available programs are listed within the reagent table of the Supplementary Information.

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

## Acknowledgements
We are grateful to Stephane Laporte (McGill University, Canada) for providing the HEK293SL cell line and to Asaka Inoue (Tohoku University, Japan) for providing the HEK293T-Gα_s KO cell line. We thank Stéphane

Laporte and Michel Bouvier (IRIC, Canada) for sharing various BRET biosensors. Our appreciation also goes to Stephane Lefrançois for the pmRFP-C3-Rab7 construct, Michel Bouvier and Roshanak Irranajad (UCSF, USA) for the SNAP-tagged $V_2R$ and $\beta_2AR$, respectively. We thank Maggie Vallières and Erwan Lanchec for their help generating several constructs and Louis-Philippe Picard and Haoran Geng for their help with BRET interpretation and limitation. We would also like to express our gratitude to Richard Leduc for his valuable insights on this article. We would like to acknowledge the Plateforme de microscopie photonique de l'Université de Sherbrooke for their support. This research was funded by the Natural Sciences and Engineering Research Council of Canada (NSERC) to Christine Lavoie (grant #RGPIN-2020-06468). Additionally, Andréanne Laniel received predoctoral fellowships from the FRQNT-funded PROTEO Network and the Fonds de Recherche du Québec (FRQS).

## Author contributions

A.L. planned and performed most of the experiments, collected and analyzed the data, made most of the figures, and drafted the manuscript. B.H. performed all BRET experiments, made the BRET figures and revised the manuscript. EL performed the BRET assay with endogenous GPCR and SJG performed WB analysis. P.-L.B. provided intellectual feedback and revised the manuscript and C.L. designed the study, provided intellectual feedback, participated in interpretation of data, and revised the manuscript. The manuscript was written through contributions of all authors. All authors have given approval to the final version of the manuscript.

## Competing interests

The authors declare no competing interest.
