## [Transparent Peer Review file · Communications Biology]

Class A and B GPCRs trigger rapid G α s translocation to late and slow recycling endosomes

Corresponding Author: Dr Christine Lavoie

Version 0:

Reviewer comments:

Reviewer #1

(Remarks to the Author)

In this very interesting manuscript from Laniel et al. the authors explore endosomal trafficking of G α s following stimulation of a class A (β 2AR) and a class B (V2R) GPCR in an heterologous HEK293 cellular model, using qualitative confocal microscopy combined with quantitative enhanced bystander BRET (ebBRET) approaches. The authors demonstrated that following stimulation of either β 2AR or V2R, G α s trafficking is independent from the receptors trafficking from the plasma membrane to endosomes. Laniel et al. observed that G α s primarily traffics to late and slow recycling endosomes, and to a smaller extent to early and fast recycling endosomes. This contrasts with β 2AR or V2R trafficking which are both mainly observed to traffic to early and fast recycling endosomes. As previously shown for G α s (<https://doi.org/10.1038/s42003-024-06512-y>) and G α q (<https://doi.org/10.1073/pnas.2025846118>), Laniel et al. confirmed that while receptor activation is essential for G protein trafficking, G α s and the activated receptors do not traffic together. The authors used a biased β 2AR agonist as well as expression of the dynamin dominant negative DynK44A which only impacted receptor trafficking to demonstrate receptor-independent G α s endosomal trafficking. Finally, the authors demonstrated the requirement of palmitoylation of G α s for its endosomal trafficking.

G protein signalling from intracellular compartments is a developing and extremely exciting area within the GPCR field. However, the mechanistic details of compartmentalised signalling are poorly understood, and increased knowledge is essential to ultimately design more directed, efficacious, and targeted therapeutics. In contrast to many studies which focus on the trafficking of GPCRs and G proteins to early endosomes, this study explores G protein trafficking to other endosomal compartments (late and recycling endosomes).

Here are a few constructive recommendations which would strengthen the importance and impact of this investigation. To be noted that the major concerns can be easily addressed by the authors.

Major concerns:

- A weakness of this study is the exclusive use of a heterologous cellular model (HEK293 cells) in which the G proteins and receptors are overexpressed. HEK293 cells are extremely useful to explore quickly molecular and signalling mechanisms as they are easy to transfect and they grow fast. However, the trafficking properties of GPCRs and/or G proteins may be different within a system in which these proteins are expressed at endogenous levels. The relevance of this study would be greatly enhanced if the authors could validate the key findings by monitoring endogenous G α s trafficking within an alternative cell type expressing the receptors endogenously.

- The authors demonstrate the trafficking properties of G α s and the receptors (β 2AR and V2R) without assessing their activation states at the various types of endosomes. Monitoring the location of both G α s and receptor states using confocal microscopy and ebBRET would be relatively easy using nanobody 37 and mini-Gs as respective probes. This may answer some questions raised by the authors in the discussion.

- Several data represent only one individual experiment (with SD) while the authors mention that each type of experiment was performed 3 times. Sometimes the number of individual experiments are also not reported in the legend. Although I understand that sometimes there is an important variation of the absolute values of BRET between experiments, upon normalization, the normalized values should be homogenous and allow to calculate an average with a reasonable SEM. If

the authors state that all 3 experiments were similar, why not show the average of normalised BRET values (Δ BRET) with SEM instead of a representative experiment with SD? Moreover, the extent of colocalization was not shown in several confocal experiments (figures 1, 6, 7, 9, and supplementary figure 2) . As confocal imaging is very subjective by nature, quantifying the images helps to increase the quantitative nature from the qualitative outputs.

- Lines 113-117 reporting to supplementary figure 2: The authors state that overexpression of $G\beta 1\gamma 1$ did not alter $G\alpha s$ trafficking. The receptor used to support this claim was $\beta 2AR$, which is a class A GPCR in regard to its relatively low affinity for β -arrestins. However, $G\beta\gamma$ is reported to alter $G\alpha s$ trafficking of class B GPCRs, which have a high affinity for β -arrestins. Consequently, to claim that $G\beta 1\gamma 1$ overexpression did not alter $G\alpha s$ trafficking, the authors should test V2R instead of $\beta 2AR$.

- Figure 4B: As $G\beta\gamma$ has been reported to increase $G\alpha s$ trafficking to early endosomes upon stimulation of V2R (<https://doi.org/10.1038/s42003-024-06512-y>), it would be interesting to see the effect of $G\beta\gamma$ overexpression or scavenging on $G\alpha s$ trafficking to the different endosomal compartments. Would modulating $G\beta\gamma$ levels have the same effect on $G\alpha s$ trafficking to other endosomes?

- Figure 5D: Comparing the agonist-mediated Δ BRET values for different BRET acceptors needs appropriate controls if using ebBRET because the relationship between BRET values and the GFP/Rluc ratio is different between BRET and ebBRET. In conventional BRET assays, luciferase from *Renilla reniformis* and GFP variants from *Aequorea victoria* do not interact spontaneously and limit non-specific signals occurring from random interactions. In this situation, the BRET values saturate when a given GFP/Rluc ratio is reached. Therefore, when using conventional BRET, the comparison of the Δ BRET values with different BRET acceptors is possible as long as the system is at saturation (enough BRET acceptor for BRET donor), and this usually requires very low ratiometric levels of BRET acceptor/BRET donor. In contrast, when using ebBRET, the BRET values do not saturate with increasing rGFP/Rluc values, but instead increase in a linear fashion until the cells cannot express any more energy donor. Therefore, it is imperative when comparing the Δ BRET values between different BRET acceptors (here rGFP-Rabs) to ensure very similar rGFP/Rluc ratios between all conditions. This ensures that each rGFP-Rab is expressed at the same levels compared to each other, and same thing for the energy acceptor (Gs-RlucII). Otherwise, the Δ BRET values will simply reflect the extent of expression of each rGFP-Rab protein. By experience, rGFP-Rab5 is more difficult to express than any other Rab protein, which would lead to the incorrect conclusion of a low Δ BRET values with Rab5...which is exactly what is observed here. While the Rluc values can be directly obtained from the ebBRET experiments, the levels of expression of the different rGFP-Rabs have to be previously monitored by stimulating directly the rGFP (not via luciferase stimulation) before performing the ebBRET experiment. Consequently, I would request the authors to report the expression of each ebBRET acceptor as well as the ebBRET donor to ensure solid conclusions. They must be equal in order to compare the Δ BRET values between different ebBRET acceptors.

- Line 177 and supplementary figures 3-5: The authors state that colocalization analysis and quantification revealed that both $\beta 2AR$ and V2R rapidly colocalize with Rab4 and Rab5, while exhibiting minimal colocalization with Rab7 and Rab11 when referring to supplementary figures 3-5. This observation is surprising because $\beta 2AR$ and V2R are the two prototypical class A and class B GPCRs in term of their relative affinities for β -arrestins. The trafficking of these two receptors has been reported many times as different ([doi: 10.1074/jbc.274.45.32248](https://doi.org/10.1074/jbc.274.45.32248)). Based on the published work, $\beta 2AR$ should spend minimal time at early endosomes (Rab5) and be recycled quickly (Rab4/11), while V2R should be minimally recycled but spend an important amount of time at the early endosomes (Rab5) before being targeted to late endosomes (Rab7) en route for lysosomal degradation. Any possible explanation as to why there is this major discrepancy compared to the literature? The absence of $\beta 2AR$ or V2R found in late (Rab7) and slow recycling (Rab11) endosomes is also mentioned in the discussion (line 292), but is not discussed. This element should be discussed as it is rather surprising.

- Supplementary figure 6: Was the relative expression of $G\alpha s$, myr- $G\alpha s$ and $G\alpha s$ -C3S the same? Was it monitored? To compare the downstream effect (cAMP production), the different versions of $G\alpha s$ must be expressed at similar levels. Also, any idea why in absence of $G\alpha s$ or in presence of myr- $G\alpha s$, the forskolin-induced stimulation is lower than in presence of wild-type $G\alpha s$ or $G\alpha s$ -C3S? When the authors state (line 234) that a lower baseline BRET signal was observed, indicating elevated basal cAMP levels and suggesting constitutive activity referring to supplementary figure 6, cautious interpretations of these results are important. Unimolecular sensors such as the EPAC construct used to measure cAMP become saturated at a certain level of cAMP. Once the sensor is completely opened (minimal BRET), increasing cAMP concentrations will not be detected as the maximally open conformation has been reached by this sensor. This may be what happens for cells expressing myr- $G\alpha s$. The already high basal cAMP levels may mask further observable cAMP produced by forskolin and iso stimulation.

Minor concerns:

- Figure 1: The yellow arrows on the merged images mask the actual colocalization as the colocalized regions are also yellow. Therefore, a different colour, or even white, may be more suitable visually.

- Figure 1D: An alternative image could be provided which may show better representation of $G\alpha s$ as the image selected shows less fluorescent intensity of the green channel compared to the other conditions and therefore transfection may not be optimal within this particular cell.

- Panel A of figure 5, and panels A and D of figure 8 were likely created using "biorender" and this should be acknowledged within the manuscript.

- Figure 5: Although I think this is simply a typo, is the rGFP fusion to Rab4 and Rab11 and the GFP10 fusion to Rab7 and

Rab5 a mistake? The Δ BRET values are much higher when rGFP is used in comparison to GFP10 (<https://doi.org/10.1038/ncomms12178>). So, to compare the Δ BRET values obtained from different Rabs, the BRET acceptor must be the same. If GFP10 is used for Rab5, there will be an artefactual under-representation of the agonist-mediated BRET compared to what is observed with rGFP-Rab11.

- Figure 8: Similarly, on the cartoons (panels A and D), CAAX and Rab7 are fused to rGFP while in panel E and F Rab7 seems to be fused to GFP10. Is it rGFP or GFP10? Also, why is only the trafficking from the plasma membrane to late endosomes are shown? Was this experiment also performed also with Rab5 and Rab11?

- Supplementary figure 7: Again, was Rab7 fused to rGFP or GFP10? There is a discrepancy between the figure legend indicating rGFP-Rab7 and the title of panel B which indicates GFP10-Rab7.

- The BRET assays described are more specifically termed as enhanced bystander BRET (ebBRET) if the naturally interacting chromophores used were luciferase and GFP from the same species (*Renilla reniformis*) (<https://doi.org/10.1038/ncomms12178>). If this is the case, please replace BRET by ebBRET in the text.

- Line 293 in discussion: The authors state that a recent study has demonstrated that the $G\beta\gamma$ dimer is predominantly localized in late endosomes, as evidenced by its colocalization with Rab7 and that this finding suggests that the $G\alpha$ subunit may also be present in these compartments, aligning with their results. According to a recent study from Nevin Lambert (<https://doi.org/10.7554/eLife.97033.2>), G proteins are already present on early, late, and recycling endosomes and receptor activation does not change their endosomal abundance. Consequently, although the authors observed only weak $G\alpha$ s trafficking to early endosomes upon stimulation of β 2AR and V2R, this observation is not incompatible with $G\alpha$ s signalling from early endosomes by these receptors if a pool of $G\alpha$ s is already present at that compartment. The important $G\alpha$ s for endosomal signalling may not be the small population previously activated by the receptors at the plasma membrane. I think this should be part of the discussion as it offers at least a path towards a possible explanation to reconcile the lack of $G\alpha$ s trafficking to early endosomes with the abundant reports of $G\alpha$ s signalling from early endosomes. The authors indeed observed a robust colocalization of β 2AR and V2R at early endosomes.

Reviewer #2

(Remarks to the Author)

Review Communication Biology 2025.

Lavoie & AI

Many GPCRs are known to couple and activate GalphaS proteins to trigger downstream cAMP signaling. In the last decade, several studies have highlighted that Gs is activated not only at the plasma membrane (PM) but also from endomembranes, especially early endosomes. The mechanism of GalphaS trafficking to endosomes for signaling is not yet fully understood.

Previous studies have shown that upon receptor stimulation, GalphaS is translocated from the PM to the cytoplasm and several endomembranes structures. However, GalphaS receptor-mediated trafficking, intracellular distribution and role in endosomal signaling by GPCR remain to be further studied.

In the proposed study, authors aim to study how GalphaS rapidly redistributes upon activation by two GPCRs known to mediate Gs-activation from early endosomes, the 2AR and V2R, and several ligands. They investigate how GalphaS trafficking correlates with receptor internalization. Using confocal microscopy and BRET approaches, authors demonstrated most GalphaS accumulate in Rab7 and Rab11 positive endosomes. 2AR and V2R seems to induce similar GalphaS trafficking.

Their data indicate the need for an intact GalphaS palmitoylation site for efficient trafficking to endomembranes but this trafficking does not require receptor endocytosis or the dynamin-dependent-endocytic machinery.

In addition, authors show low and/or transient localization with receptors and more prolonged residence in Rab5+ early endosomes and Rab4+ structures.

So far, the manuscript is well written and easy to read. Main findings highlighted by this study emphasize the essential role of GalphaS activation by agonists at the PM for trafficking and add incremental information that should be of interest for the GPCR community.

Specific comments:

- Confocal imaging was used to investigate the translocation of GalphaS from the PM to endocytic vesicles. However, in most confocal images except for Fig. 2 & 9, GalphaS and receptors do not appear rather as punctae than at the membrane. A cell surface labelling delineating the cell membrane is usually expected. It is especially expected for GPCRs such as V2R or 2AR well described in the literature. Therefore, in this experiment set up, it is unclear whether upon agonist stimulation, GalphaS redistribution occurs essentially from PM to vesicles or whether cytoplasmic location portion is also targeted to vesicles. Authors should explain this and show images with optimized cell surface localization of GalphaS and receptors as in J.A Allen & AI, Mol Pharm 2005 (10.1124/mol.104.008342) and Irranejad & AI, Nature 2013 (10.1038/nature12000) respectively.

- For BRET assay, GalphaS spatio-temporal trafficking was done using two different GalphaS isoforms construct (short and long isoforms) but they were not compared to each other. This raises the question whether both isoforms equal regarding

their cellular distribution upon GPCR activation. Could authors comment on this?

- Although authors have shown that various agonists with functional selectivity are able to induce GalphaS trafficking, in the current version of the manuscript, saturating concentration of agonist were used. However it remain elusive whether GalphaS trafficking is dose dependent. This is important considering that GalphaS may translocate only under certain condition of stimulation and/or receptor expression level.

Reviewer #3

(Remarks to the Author)

Summary:

The manuscript by Laniel et al. aims to explore the distribution of G α s to subcellular compartments following stimulation of their activating GPCRs (i.e., prototypical Class A receptor adrenergic β 2AR and Class B receptor V2R). Overexpression of labeled G α s, namely G α s-GFP and G α s-RLucII, for live-cell confocal microscopy and BRET, respectively, is the primary method to quantitatively assess distribution of G α s for this study. Consistent with prior publications, Laniel et al. shows that G α s translocate away from the plasma membrane (PM). Specific subcellular compartments, namely Rab7+ compartment and Rab11+ compartments thought to represent late endosomes and slow recycling endosomes, respectively, are the preferred destinations of G α s following stimulation. These sites are different from preferred destination for activated β 2AR and V2R as these receptors tend to aggregate in early endosomes (Rab5) and rapid recycling endosomes (Rab4). This finding raises the question of how GPCRs signal from endosomes if the predominant destinations of activated receptors and heterotrimeric G proteins differ. Additional findings include how modification of palmitoylation site of G α s leads to disruption of G α s translocation despite preserved its activation.

Major impressions:

1. Given that there has been great interest in studying the role of GPCR-G protein endosomal signaling, Laniel et al.'s assessment of heterotrimer G protein availability in endosomal compartments is an important question. Specifically, this study challenges the commonly overlooked assumption that G protein heterotrimers are always present at the location where activated GPCRs redistribute. Overall, the experiments are well designed and utilizing two different experimental approaches (i.e., BRET and fluorescent microscopy) bolster the conclusion of the findings. However, there are some discrepancies in confocal data such as the clear differences in GFP signal in cytoplasm and distribution of unstimulated receptors (Please refer to Comments 1 and 2).
2. Given much of the findings had been either done or inferred from prior works reduces the novelty of this study. For example, the observation that G α s dissociates from the PM upon activation has been extensively studied and known for years as the authors have already pointed out. The use of bystander BRET to study the kinetics and steady state of G α s-RLuc translocation to intracellular compartments (endoplasmic reticulum, recycling endosomes, late endosomes) has been already reported by Martin and Lambert 2016; PMID: 27528603, albeit there are discrepancies in interpretation including the role of palmitoylation.
3. The title of the manuscript generalizes the findings to Class A and Class B GPCRs but only one receptor representing each class was studied. I believe it would be misrepresentative to generalize the finding. In addition, the translocation occurring around two minutes should not be described as rapid. In the context of cellular signaling (i.e., cAMP in this case), scale of minutes would not be the scale I would expect for rapid.

Comments:

1. The description that there is a fair amount of G α s in the cytoplasm (Line 97-99) does not appear consistent with prior studies describing the distribution of G $\beta\gamma$ subunits utilizing CRISPR-mediated gene editing (Cho et al., 2022 PMID: 35271311 and Jang et al., 2024 PMID: 39514269). I have a suspicion that this cytoplasm heavy distribution is secondary to overexpression as there are previous reports suggesting inability for free G α or G $\beta\gamma$ subunits to efficiently traffic to plasma membrane (Wedegaertner, 2012 PMID: 23161140). There is also discrepancy in relative PM to cytoplasm ratio for 0 min G α s-GFP in figure 2A and 2D compared to that of Figure 1. I would recommend doing control experiments looking at relationship between expression of G α s-GFP/RLuc and spatial distribution. Easiest and least-descriptive experiment would be to do bystander BRET for G α s-RLuc. Total luminescence can be counted for overall expression amount. Moreover, there should be a quantitative assessment of G α s-GFP distribution, perhaps by line analysis.
2. It is unclear to me why receptor expression is so punctate at baseline for V2R in Figure 1B. Can authors explain this?
3. For accurate assessment of GPCR localization, it would be important to know that the majority of receptors are indeed labeled and we are not heavily underlabeling and missing population of receptors that we cannot observe. I suggest authors to demonstrate the efficiency of labeling tag-GPCR constructs by expression tag-GPCR-Fluorescent Tag labeled by antibody in different concentrations.
4. Authors point that Figure 1C-1D demonstrates that G α s trafficking is dependent on activation of G α s-activating GPCRs. However, there are no control experiments showing that the ligands used for CXCR4 (representative of Gi) and AT1R (representative of Gq and Gi) are indeed functional since there are no positive controls. I personally do not believe Fig 1C and 1D are needed since it's intuitive that Gs wouldn't be affected. However, if authors choose to address this question, I recommend using GFP-mini G proteins (GFP-mGsi/mGsqs) or GFP-arrestin.
5. It is sub-optimal that authors conducted separate imaging experiments for G α s-Rab-GPCR colocalization. It likely would have been more flexible to use SNAPf-GPCR for flexibility in avoiding wavelength overlap.
6. To address whether the size of GFP tag affected G α s trafficking (Fig S1a), authors addressed that the limitation of this

approach is high level of GFP signal in cytoplasm due to frankenbody-GFP that is not bound to G α s-HA. I would recommend trying split fluorescent protein strategy with mNG2 (11) tag attached to G α s and cytoplasmic expression of mNG2 (1-10). The advantage would be that mNG2 construct is mostly fluorogenic only when they are complemented.

7. For assessment of whether G $\beta\gamma$ overexpression also affects the distribution G α s, it is unusual that the authors focused solely on G β 1 γ 1 since farnesylation of G γ 1 affects the distribution and kinetics of dimer translocation upon activation (Kankanamge et al., 2022 PMID: 36272647). Namely, farnesylated gamma subunits (1,9,10) are more readily distributed away from PM upon activation. Because the confocal data did not suggest change in distribution for this condition, I do not believe testing other gamma subunits is necessary. However, this should be addressed. If authors choose to keep the G $\beta\gamma$ overexpression data, there should be control assessing whether G β 1 γ 1 is indeed the predominant dimer forming complexes with G α s-GFP.

8. The fact that there is affinity (perhaps in micromolar scale) between rGFP and RlucII should be mentioned (Line 163-165) per Molinari et al., 2008 (PMID: 17868039) and Namkung 2016 (PMID: 27397672). While the affinity is strong enough to bias the system towards intracellular redistribution is questionable, it should nevertheless be mentioned. Furthermore, the affinity is even more unclear in the context of cytoplasm to membrane or membrane to membrane, that is 3D to 2D and 2D to 2D interactions, respectively.

9. Please show baseline BRET for steady state experiments (Figure 5D). While the baseline number is impacted by conformation and not entirely reliable for quantitative purpose, clear differences especially in comparison between different GPCR overexpression may provide additional information. Plus, also useful for assessing validity of the experiments.

10. It would be ideal to quantify kinetics for Figure 8B.

11. What accounts for baseline BRET difference (much lower in fig. S7 around 0.24 than fig. 8B 0.4)?

12. Period at the end of paragraph Line 184.

13. Was the Dyn K44A experiment done in the context of using bystander BRET? If so, did it line up with the confocal findings?

14. In Line 365-367, authors claim that the G α s endosomal translocation is independent of β -arrestin recruitment but this was not experimentally addressed by authors. Please either conduct the appropriate experiments or cite other studies that have suggested it (Kwon et al., 2022 PMID: 36289326).

Version 1:

Reviewer comments:

Reviewer #1

(Remarks to the Author)

Laniel et al. have done an excellent job addressing most of the concerns previously raised. As a result, the findings presented in the manuscript are now significantly more robust and convincing. Below are a few remaining concerns that should be readily addressable:

1- I remain concerned about the lack of effect of G $\beta\gamma$ on G α s trafficking. I wonder whether the use of G α s119-RLucII in the present manuscript (rather than G α s67-RLucII as used by Sokrat et al.) masks the impact of G $\beta\gamma$ overexpression on G α s trafficking. Could the authors test the effect of G $\beta\gamma$ overexpression on G α s67-RLucII trafficking? FYVE and Rab5 are both well-established markers of early endosomes making the G α s119-RLucII construct the most plausible explanation for the discrepancy between the present study and that of Sokrat et al. This experiment would be crucial to ensure the results are not artefactual (especially given that the findings appear to contradict a pivotal concept recently published by the Bouvier group).

2- The new panels 6F and 6I are very interesting as they demonstrate that G α s translocation from the plasma membrane to late endosomes is concentration-dependent. While the authors state that similar potencies are observed for each ligand, there are clearly noticeable differences that should not be overlooked. In particular, the potency of dobutamine appears significantly lower than that of isoproterenol, salbutamol, and epinephrine. This should be acknowledged and if possible briefly discussed with a possible explanation.

Reviewer #2

(Remarks to the Author)

Laniel & al aimed at investigated the spatio-temporal distribution of GalphaS upon two prototypical Class A & class B GPCRs, the beta2-adrenergic receptor and the arginine-vasopressin receptor type 2, respectively. Authors added new data that clearly strengthen and support the observation that GalphaS is redistributed intracellularly with an apparent enrichment on Rab7 and Rab11 positive endosomes and extended these observation with two additional representative GPCRs of class A & class B. This work further add to better understand how G proteins (here GalphaS) are regulated beyond activation and dissociation from Gbetagamma. Nonetheless, further work would be needed to understand a possible function of GalphaS in the different rab7 and Rab11-endosomes.

In my opinion, Laniel & al have answered most criticisms and comments raised upon the initial round of review and in my opinion, the revised manuscript is convincing and is now compatible with publication.

Reviewer #3

(Remarks to the Author)

Major Impression 2:

While I agree that these are indeed the differences in experiment design between the current study and prior studies, I am not sure whether the current study fundamentally changes our understanding of activated $G_{\alpha s}$ that we suspect do sample different endosomal compartments. And without monitoring endogenous G_{α} subunits (which no group has published at this point of writing), it is difficult to say whether overexpression-based methods make substantial progress on this issue. The findings would be limited to the fact that engineered tools are used to postulate how the endogenous protein would behave and differences noted here are directly compared to prior tools. I am not sure how using eBRET over BRET1 was necessary to demonstrate the findings as well.

Major Impression 3:

While the timing of what is "rapid" is relative to what process we are comparing to, I still believe we cannot generalize the current studies to Class A and Class B GPCRs based on two receptors for each category. There is no specific number that meets the cutoff of what makes a certain study generalizable to the entire class but studying two receptors per class is on the smaller scale, especially considering BRET-based assays are pretty scalable. While I do not think it is fundamental for the authors to study more receptors to prove their points, it would be misleading to state the title as it is. Please consider removing the context of Class A and Class B GPCRs and rather emphasize what makes $G_{\alpha s}$ signaling different.

Comment 1:

It is recommended to explicitly state that the apparent cytoplasmic distribution of $G_{\alpha s}$ is likely due to overexpression as there is currently no available data of endogenous $G_{\alpha s}$ distribution in live cells. The cell-to-cell variability should be mentioned in results/discussion, not just methods as this is an important point for interpretation. Transfection at low level helps but does not resolve the issue because either/or the amount of plasmid delivered per cell and efficiency of expression are variable processes that are difficult to be titrated.

Comments 2-5:

Authors have addressed this question.

Comment 6:

While authors do not have to conduct the suggested study, I believe it is important to mention that it should not be assumed that $G_{\alpha s}$ -HA has similar trafficking as wildtype $G_{\alpha s}$ just because it has a smaller tag.

Comments 7-14:

Authors have addressed this question.

Additional correction:

Please include subscripts for receptors (2 for $\beta 2AR$, $V2R$) and G proteins (s for G_s , $G_{\alpha s}$)

Line 15: "Heterotrimeric $G_{\alpha s}$ is classically known for mediating G protein-coupled receptor (GPCR)"
Since $G_{\alpha s}$ is monomeric, would say " $G_{\alpha s}$ is classically known...". Or could say heterotrimeric G_s

Please find enclosed our revised version of manuscript ID COMMSBIO-24-8266 by Laniel et al., initially entitled "*Stimulation of class A and class B GPCRs leads to rapid translocation of Gas to late and slow recycling endosome*". We wish to thank the reviewers for their helpful suggestions, which have improved the manuscript. We believe that we have adequately addressed all the issues that were raised. Below, you will find our detailed responses to the reviewers' comments.

Reviewer #1

Major concerns:

1) A weakness of this study is the exclusive use of a heterologous cellular model (HEK293 cells) in which the G proteins and receptors are overexpressed. HEK293 cells are extremely useful to explore quickly molecular and signalling mechanisms as they are easy to transfect and they grow fast. However, the trafficking properties of GPCRs and/or G proteins may be different within a system in which these proteins are expressed at endogenous levels. The relevance of this study would be greatly enhanced if the authors could validate the key findings by monitoring endogenous Gas trafficking within an alternative cell type expressing the receptors endogenously.

We agree that validation in a more physiologically relevant context would strengthen the impact of our findings. To address this concern, we have now extended our study to include experiments with β 2-adrenergic receptor endogenously express in HEK293 cells (see section "*Gas endosomal redistribution induced by other class A and B GPCRs and endogenous GPCR*", page 6-7 of the revised manuscript). Using ebBRET methodology, we monitored Gas trafficking in response to endogenous β 2AR activation. The results, now presented in Figure 5C, demonstrate that Gas exhibits similar trafficking patterns as those observed in the overexpression system. This consistency confirms that our key findings are not limited to heterologous expression models but also apply under conditions of endogenous receptor expression.

2) The authors demonstrate the trafficking properties of Gas and the receptors (β 2AR and V2R) without assessing their activation states at the various types of endosomes. Monitoring the location of both Gas and receptor states using confocal microscopy and ebBRET would be relatively easy using nanobody 37 and mini-Gas as respective probes. This may answer some questions raised by the authors in the discussion.

We agree that assessing the activation states of Gas and the receptor at different endosomal compartments is an important question. However, this is beyond the scope of the current study and will be the focus of our future work.

3) Several data represent only one individual experiment (with SD) while the authors mention that each type of experiment was performed 3 times. Sometimes the number of individual experiments are also not reported in the legend. Although I understand that sometimes there is an important variation of the absolute values of BRET between experiments, upon normalization, the normalized values should be homogenous and allow to calculate an average with a reasonable SEM. If the authors state that all 3 experiments were similar, why not show the average of normalised BRET values (Δ BRET) with SEM instead of a representative experiment with SD? Moreover, the extent of colocalization was not shown in several confocal experiments (figures 1, 6, 7, 9, and supplementary figure 2). As confocal imaging is very subjective by nature, quantifying the images helps to increase the quantitative nature from the qualitative outputs.

As recommended, we have now normalized the majority of BRET experiments to Δ BRET and calculated the average with SEM. Exceptions include Figures 8B and Supplementary Figure 9A, where baseline values are shown for proper data interpretation, as well as Supplementary Figures 2 and 7, which display baseline values at the request of reviewer #3 to complement the Δ BRET data. Additionally, all figure legends have been updated to clearly indicate the number of independent experiments conducted for each figure.

We agree that confocal imaging can be subjective, and that quantitative analysis is often necessary to support conclusions drawn from confocal imaging. In the case of Figure 1, we did perform colocalization quantification (see below); however, we found that the results primarily reflect an initially high level of colocalization at the plasma membrane, which gradually decreases over time, ultimately yielding a low Manders coefficient indicative of minimal or negligible colocalization. This trend mirrors the overall distribution pattern visible in the confocal images and does not provide additional information, particularly

given the absence of discrete vesicular colocalization. As such, we do not believe that including this quantification in the manuscript would enhance data interpretation.

For Figures 6 (various agonist stimulation), Figure 7 (effect of DynK44A), Figure 9 (now Fig. 8A, impact of Gas-Myr and Gas-C3S), and Supplementary Figure 2 (now Suppl. Fig. 6, G β γ overexpression), the purpose of the confocal imaging was to determine whether these conditions alter the translocation of Gas on intracellular vesicles rather than to specifically quantify colocalization with receptors. In each case, our qualitative imaging observations are supported by complementary quantitative BRET assays, which provide robust confirmation of our interpretations. Given the specific objectives of these experiments and the supporting quantitative BRET data, we believe that additional, resource-intensive colocalization analyses would not substantially advance the understanding or interpretation of our findings.

4) Lines 113-117 reporting to supplementary figure 2: The authors state that overexpression of G β 1 γ 1 did not alter Gas trafficking. The receptor used to support this claim was β 2AR, which is a class A GPCR in regard to its relatively low affinity for β -arrestins. However, G β γ is reported to alter Gas trafficking of class B GPCRs, which have a high affinity for β -arrestins. Consequently, to claim that G β 1 γ 1 overexpression did not alter Gas trafficking, the authors should test V2R instead of β 2AR.

We agree that, given the reported influence of G β γ on Gas trafficking through class B GPCRs, it is more appropriate to assess this effect using V2R rather than β 2AR. To address this, we conducted additional experiments in HEK293 cells overexpressing V2R and G β 1 γ 2, the appropriate G β γ dimer for V2R and Gas. Upon V2R stimulation, we found that G β 1 γ 2 overexpression did not affect Gas trafficking to vesicles, as assessed by confocal microscopy, nor did it significantly impact Gas trafficking to all Rab-labeled endosomal compartments in our ebBRET-based assay (see Suppl. Figure 6). These results indicate that, in our system, G β 1 γ 2 overexpression does not alter Gas trafficking following V2R activation. We note that this finding contrasts with a previous report (Sokrat et al., 2024, PMID: 38972875), which described increased Gas recruitment to endosomes upon G β γ overexpression. This discrepancy may be explained by differences in the methodology (we did not use G β γ scavenger tools) or in the biosensors used, as our study employed Gas119-RLucII and Rab5-rGFP, whereas Sokrat et al. used Gas67-RLucII and rGFP-FYVE.

5) Figure 4B: As G β γ has been reported to increase Gas trafficking to early endosomes upon stimulation of V2R (<https://doi.org/10.1038/s42003-024-06512-y>), it would be interesting to see the effect of G β γ overexpression or scavenging on Gas trafficking to the different endosomal compartments. Would modulating G β γ levels have the same effect on Gas trafficking to other endosomes?

These experiments have been performed (see Suppl Fig 6), as described in our response to comment #4.

6) Figure 5D: Comparing the agonist-mediated Δ BRET values for different BRET acceptors needs appropriate controls if using ebBRET because the relationship between BRET values and the GFP/Rluc ratio is different between BRET and ebBRET. In conventional BRET assays, luciferase from *Renilla reniformis* and GFP variants from *Aequorea victoria* do not interact spontaneously and limit non-specific signals occurring from random interactions. In this situation, the BRET values saturate when a given GFP/Rluc ratio is reached. Therefore, when using conventional BRET, the comparison of the Δ BRET values with different BRET acceptors is possible as long as the system is at saturation (enough BRET acceptor for BRET donor), and this usually requires very low ratiometric levels of BRET acceptor/BRET donor. In contrast, when using ebBRET, the BRET values do not saturate with increasing rGFP/Rluc values, but

instead increase in a linear fashion until the cells cannot express any more energy donor. Therefore, it is imperative when comparing the Δ BRET values between different BRET acceptors (here rGFP-Rabs) to ensure very similar rGFP/RLuc ratios between all conditions. This ensures that each rGFP-Rab is expressed at the same levels compared to each other, and same thing for the energy acceptor (Gs-RLucII). Otherwise, the Δ BRET values will simply reflect the extent of expression of each rGFP-Rab protein. By experience, rGFP-Rab5 is more difficult to express than any other Rab protein, which would lead to the incorrect conclusion of a low Δ BRET values with Rab5...which is exactly what is observed here. While the RLuc values can be directly obtained from the ebBRET experiments, the levels of expression of the different rGFP-Rabs have to be previously monitored by stimulating directly the rGFP (not via luciferase stimulation) before performing the ebBRET experiment. Consequently, I would request the authors to report the expression of each ebBRET acceptor as well as the ebBRET donor to ensure solid conclusions. They must be equal in order to compare the Δ BRET values between different ebBRET acceptors.

The reviewer is correct to point out the discrepancy between GFP10 and rGFP acceptors used in earlier versions of the manuscript. We would like to clarify that our intent was never to quantitatively compare different Rab biosensors. It is widely recognized that distinct BRET pairs cannot be directly compared, as factors such as orientation and intracellular localization may alter BRET efficiency even at similar expression levels. Our primary goal was to analyze qualitative changes within each biosensor system. In response to the reviewer's comments, we have revised the manuscript to standardize all Rab BRET assays by exclusively using tandem repeat rGFP-Rab constructs (Tdr-rGFP-Rabs). We also systematically quantified rGFP fluorescence to confirm that acceptor expression levels are comparable across all conditions thus ensuring consistent parameters for ebBRET measurements. As shown in the graphs below, acceptor fluorescence quantification demonstrated that rGFP-Rab expression was similar and not significantly different between conditions. Additionally, we assessed donor RLucII levels and observed some variability, with lower levels primarily when Gas was co-expressed with Rab7 or Rab11. As reported by Namkung et al. (Nat Commun 2016, PMID: 27397672), ebBRET is a highly efficient energy transfer system. Therefore, these lower luminescence values may reflect increased basal interaction—particularly between Gas and Rab11 compared to Gas and Rab5—rather than differences in expression levels. In line with this interpretation, we consistently observed higher baseline BRET ratios with rGFP-Rab11 than with rGFP-Rab5, as shown in Supplementary Figures 2 and 7. For these reasons, we do not consider RLucII counts to be reliable indicators of expression in this context. In summary, we have revised the manuscript to eliminate direct quantitative comparisons between different Rab biosensors and have clarified these points in the revised text (see page 5-6). We were not planning to add this data to the manuscript, but we can do it if required by the reviewer.

7) Line 177 and supplementary figures 3-5: The authors state that colocalization analysis and quantification revealed that both β 2AR and V2R rapidly colocalize with Rab4 and Rab5, while exhibiting minimal colocalization with Rab7 and Rab11 when referring to supplementary figures 3-5. This observation is surprising because β 2AR and V2R are the two prototypical class A and class B GPCRs in term of their

relative affinities for β -arrestins. The trafficking of these two receptors has been reported many times as different (doi: 10.1074/jbc.274.45.32248). Based on the published work, β 2AR should spend minimal time at early endosomes (Rab5) and be recycled quickly (Rab4/11), while V2R should be minimally recycled but spend an important amount of time at the early endosomes (Rab5) before being targeted to late endosomes (Rab7) en route for lysosomal degradation. Any possible explanation as to why there is this major discrepancy compared to the literature? The absence of β 2AR or V2R found in late (Rab7) and slow recycling (Rab11) endosomes is also mentioned in the discussion (line 292), but is not discussed. This element should be discussed as it is rather surprising.

Thank you for drawing attention to the discrepancies between our initial colocalization results and established literature on β 2AR and V2R trafficking. We believe these differences arose from our original use of fluorescent antibody labeling of FLAG- or Myc-tagged receptors at the cell surface, which may have resulted in antibody dissociation during endocytic trafficking and thus underrepresentation of receptors in certain endosomal compartments, particularly Rab7 and Rab11. To address this, we repeated the assays using SNAP-tagged β 2AR and V2R, which allows for covalent fluorescent labeling and stable tracking of the receptors throughout their endocytic itinerary. With this improved approach, we observed trafficking patterns consistent with published data: β 2AR showed predominant colocalization with Rab5 and Rab4 early after internalization and increased Rab11 association over time, while V2R was mainly found in Rab5-positive endosomes initially and increasingly colocalized with Rab7, with little presence in Rab4 or Rab11 compartments. These new results, including microscopy images and quantification, are now included in the revised manuscript (see Supplementary Figures 3-5), and we have updated the discussion.

8) Supplementary figure 6: Was the relative expression of Gas, myr-Gas and Gas-C3S the same? Was it monitored? To compare the downstream effect (cAMP production), the different versions of Gas must be expressed at similar levels.

We have confirmed by Western blot that Gas-WT and Gas-C3S were expressed at comparable levels, ensuring that the differences observed in cAMP production are not due to variations in protein expression. While the expression level of myr-Gas was lower, which may lead to reduced cAMP response, this does not affect the main conclusion of the experiment. Our primary objective was to demonstrate that each Gas variant is functionally active and capable of stimulating adenylyl cyclase. The Western blot data have been included in the Supplementary Fig. 9B.

9) Also, any idea why in absence of Gas or in presence of myr-Gas, the forskolin-induced stimulation is lower than in presence of wild-type Gas or Gas-C3S? When the authors state (line 234) that a lower baseline BRET signal was observed, indicating elevated basal cAMP levels and suggesting constitutive activity referring to supplementary figure 6, cautious interpretations of these results are important. Unimolecular sensors such as the EPAC construct used to measure cAMP become saturated at a certain level of cAMP. Once the sensor is completely opened (minimal BRET), increasing cAMP concentrations will not be detected as the maximally open conformation has been reached by this sensor. This may be what happens for cells expressing myr-Gas. The already high basal cAMP levels may mask further observable cAMP produced by forskolin and iso stimulation.

In the absence of endogenous Gas (HEK Gas KO cells), forskolin failed to elevate cAMP levels, consistent with previous reports showing that the affinity of forskolin for adenylyl cyclase is markedly reduced without Gas, highlighting the essential role of Gas in forskolin-mediated adenylyl cyclase activation (PMID: 9268375, 9268376, 8405168, 7949691). In cells expressing myr-Gas, the forskolin-induced cAMP response (Δ BRET) appeared diminished compared to cells expressing wild-type Gas or Gas(C3S), primarily because the baseline BRET signal was already lower—reflecting elevated basal cAMP levels and suggesting constitutive activity of the myr-Gas construct, as previously reported by Michel Bouvier's group (PMID: 40136046). Importantly, despite these baseline differences, the maximal forskolin-induced BRET response with Myr-Gas reached a similar level (approximately 0.2), compared to the other constructs (Gas(C3S) can reach even lower value), indicating that the EPAC biosensor is not saturated in the presence of myr-Gas. We believe that the reduced forskolin-induced stimulation observed with Myr-Gas arises because its activity is restricted to adenylyl cyclases at the plasma membrane. Nevertheless, the primary objective of these experiments was to confirm the functionality of each Gas construct and their ability to stimulate adenylyl cyclase

Minor concerns:

10) Figure 1: The yellow arrows on the merged images mask the actual colocalization as the colocalized regions are also yellow. Therefore, a different colour, or even white, may be more suitable visually.

We have now updated all the figures to use only white arrows, which provide better visual contrast and do not obscure the colocalization signal.

11) Figure 1D: An alternative image could be provided which may show better representation of Gas as the image selected shows less fluorescent intensity of the green channel compared to the other conditions and therefore transfection may not be optimal within this particular cell.

Figure 1 C and D have been removed, as requested by Reviewer # 3

12) Panel A of figure 5, and panels A and D of figure 8 were likely created using “biorender” and this should be acknowledged within the manuscript.

We have now properly acknowledged the use of Biorender in Figures 5 and 6.

13) Figure 5: Although I think this is simply a typo, is the rGFP fusion to Rab4 and Rab11 and the GFP10 fusion to Rab7 and Rab5 a mistake? The Δ BRET values are much higher when rGFP is used in comparison to GFP10 (<https://doi.org/10.1038/ncomms12178>). So, to compare the Δ BRET values obtained from different Rabs, the BRET acceptor must be the same. If GFP10 is used for Rab5, there will be an artefactual under-representation of the agonist-mediated BRET compared to what is observed with rGFP-Rab11.

As noted in our response to comment #6, we initially used GFP10-tagged Rab5 and Rab7 alongside rGFP-tagged Rab4 and Rab11, as our intention was not to compare Δ BRET values between different Rab proteins. In the revised manuscript, however, we have standardized all Rab BRET assays by exclusively using rGFP-Rab constructs for every assay to ensure consistency. Importantly, adopting rGFP-tagged Rabs did not alter our conclusions.

14) Figure 8: Similarly, on the cartoons (panels A and D), CAAX and Rab7 are fused to rGFP while in panel E and F Rab7 seems to be fused to GFP10. Is it rGFP or GFP10? Also, why is only the trafficking from the plasma membrane to late endosomes are shown? Was this experiment also performed also with Rab5 and Rab11?

GFP10 has been replaced with rGFP in all figures, as we now exclusively use rGFP-tagged Rab constructs. The redistribution of Gas across the different Rab compartments following stimulation with the various ligands is now presented in Supplementary Figure 8.

15) Supplementary figure 7: Again, was Rab7 fused to rGFP or GFP10? There is a discrepancy between the figure legend indicating rGFP-Rab7 and the title of panel B which indicates GFP10-Rab7.

GFP10 has been replaced with rGFP in all figures, as we now exclusively use rGFP-tagged Rab constructs.

16) The BRET assays described are more specifically termed as enhanced bystander BRET (ebBRET) if the naturally interacting chromophores used were luciferase and GFP from the same species (*Renilla reniformis*) (<https://doi.org/10.1038/ncomms12178>). If this is the case, please replace BRET by ebBRET in the text.

We have revised the manuscript to consistently use "ebBRET" throughout the text wherever appropriate.

17) Line 293 in discussion: The authors state that a recent study has demonstrated that the G β dimer is predominantly localized in late endosomes, as evidenced by its colocalization with Rab7 and that this finding suggests that the G α subunit may also be present in these compartments, aligning with their results. According to a recent study from Nevin Lambert (<https://doi.org/10.7554/eLife.97033.2>), G proteins are already present on early, late, and recycling endosomes and receptor activation does not change their endosomal abundance. Consequently, although the authors observed only weak Gas trafficking to early endosomes upon stimulation of β 2AR and V2R, this observation is not incompatible with Gas signalling from early endosomes by these receptors if a pool of Gas is already present at that compartment. The important Gas for endosomal signalling may not be the small population previously activated by the receptors at the plasma membrane. I think this should be part of the discussion as it offers at least a path

towards a possible explanation to reconcile the lack of Gas trafficking to early endosomes with the abundant reports of Gas signalling from early endosomes. The authors indeed observed a robust colocalization of β 2AR and V2R at early endosomes.

Thank you for this insightful comment. We agree that the presence of a pre-existing pool of Gas on early endosomes, as shown in the study by Nevin Lambert, offers a valuable explanation for the discrepancies with our observations. This point has now been included in the discussion (page 12) to help reconcile the limited Gas trafficking we observed with the well-established role of early endosomal Gas signaling.

Reviewer #2 (Remarks to the Author):

Specific comments:

1) - Confocal imaging was used to investigate the translocation of GalphaS from the PM to endocytic vesicles. However, in most confocal images except for Fig. 2 & 9, GalphaS and receptors do not appear rather as punctate than at the membrane. A cell surface labelling delineating the cell membrane is usually expected. It is especially expected for GPCRs such as V2R or b2AR well described in the literature. Therefore, in this experiment set up, it is unclear whether upon agonist stimulation, GalphaS redistribution occurs essentially from PM to vesicles or whether cytoplasmic location portion is also targeted to vesicles. Authors should explain this and show images with optimized cell surface localization of GalphaS and receptors as in J.A Allen & AI, Mol Pharm 2005 (10.1124/mol.104.008342) and Irranejad & AI, Nature 2013 (10.1038/nature12000) respectively.

We agree with the reviewer that the images previously selected did not always clearly show plasma membrane localization of the GPCR and Gas, likely due to the use of cell-surface antibody labeling, which may have caused receptor aggregation and punctate staining. To address this artifact, we repeated the assays using SNAP-tagged receptors labeled with a fluorescent dye that covalently binds to the cell surface receptor. This approach provided a more accurate representation of plasma membrane localization and eliminated the punctate labeling observed in the earlier version of the manuscript.

2) For BRET assay, GalphaS spatio-temporal trafficking was done using two different GalphaS isoforms construct (short and long isoforms) but they were not compared to each other. This raises the question whether both isoforms equal regarding their cellular distribution upon GPCR activation. Could authors comment on this?

The reviewer is correct in noting that different Gas isoforms were used in the BRET trafficking assays. Specifically, the short isoform (Gas(119)-RLucII) was used for all assays involving Rabs, while the long isoform (Gas(67)-RLucII) was used in the CAAX assays, as it corresponds to the biosensor developed by Michel Bouvier's group. We found that the CAAX BRET assay does not produce a robust signal with Gas(119)-RLucII, necessitating the use of the Gas(67)-RLucII for these experiments. However, the reviewer raises an interesting point regarding potential differences in the cellular distribution of these isoforms upon GPCR activation. While this is beyond the scope of the current study, it represents an important question that we aim to address in future investigations.

3) Although authors have shown that various agonists with functional selectivity are able to induce GalphaS trafficking, in the current version of the manuscript, saturating concentration of agonist were used. However it remain elusive whether GalphaS trafficking is dose dependent. This is important considering that GalphaS may translocate only under certain condition of stimulation and/or receptor expression level.

The reviewer raises an important point regarding the potential dose dependency of Gas translocation. To address this, we performed concentration–response experiments using both the Gas-RLucII/rGFP-CAAX assay (plasma membrane marker) and the Gas-RLucII/rGFP-Rab7 assay (late endosome marker). The results, now presented in Figures 6F and 6I, show a clear correlation between increasing agonist concentration and Gas dissociation from the plasma membrane, along with a corresponding increase in its recruitment to late endosomes. We observed similar potencies for each ligand in both assays, as reflected by the following logEC50 values: Iso: –9.792 (CAAX), –9.499 (Rab7); Epi: –8.739 (CAAX), –8.498 (Rab7); Dob: –7.326 (CAAX), –7.063 (Rab7); and Sal: –9.392 (CAAX), –9.334 (Rab7). These findings (included in page 8 of the revised manuscript) indicate that Gas translocation to

endosomes is indeed concentration-dependent and tends to occur more readily under conditions of strong receptor stimulation.

Reviewer #3 (Remarks to the Author):

Major impressions:

1) Given that there has been great interest in studying the role of GPCR-G protein endosomal signaling, Laniel et al.'s assessment of heterotrimer G protein availability in endosomal compartments is an important question. Specifically, this study challenges the commonly overlooked assumption that G protein heterotrimers are always present at the location where activated GPCRs redistribute. Overall, the experiments are well designed and utilizing two different experimental approaches (i.e., BRET and fluorescent microscopy) bolster the conclusion of the findings. However, there are some discrepancies in confocal data such as the clear differences in GFP signal in cytoplasm and distribution of unstimulated receptors (Please refer to Comments 1 and 2).

These points are answered in Comments 1 and 2

2) Given much of the findings had been either done or inferred from prior works reduces the novelty of this study. For example, the observation that Gas dissociates from the PM upon activation has been extensively studied and known for years as the authors have already pointed out. The use of bystander BRET to study the kinetics and steady state of Gas-RLuc translocation to intracellular compartments (endoplasmic reticulum, recycling endosomes, late endosomes) has been already reported by Martin and Lambert 2016; PMID: 27528603, albeit there are discrepancies in interpretation including the role of palmitoylation.

While it is true that prior studies, including Martin and Lambert (2016), have explored Gas subcellular localization and trafficking using bystander BRET, our work advances this area in several important ways. First, we provide a detailed kinetic analysis of Gas trafficking to specific endosomal compartments in response to distinct GPCRs (Class A and class B and endogenous GPCR) and different ligands, combining both confocal imaging and an enhanced bystander BRET (ebBRET) platform. Compared to the BRET1 (RLuc8/Venus) system used previously, our use of RLucII/rGFP offers greater sensitivity and lower background (Namkung et al., 2016, PMID: 27397672), enabling us to resolve trafficking events with improved accuracy and potentially accounting for some differences in findings. Regarding the role of palmitoylation, differences in experimental tools may also explain discrepancies between studies. Martin and Lambert utilized a Gas-RLuc8 construct with RLuc8 replacing residues 73–84, and a palmitoylation-deficient mutant that also introduces a myristoylation site (so-called Myr+palm-). In contrast, we used a Gas-C3S mutant—mutated only at the palmitoylation site and bearing a RLucII insertion at residue 119. These differences in construct design can affect localization and trafficking dynamics, potentially underlying the variations seen between studies. In summary, while building on established observations, our optimized approaches and kinetic analyses provide new and complementary insights into the dynamics and mechanistic regulation of Gas trafficking that extend beyond previous work.

3) The title of the manuscript generalizes the findings to Class A and Class B GPCRs but only one receptor representing each class was studied. I believe it would be misrepresentative to generalize the finding. In addition, the translocation occurring around two minutes should not be described as rapid. In the context of cellular signaling (i.e., cAMP in this case), scale of minutes would not be the scale I would expect for rapid.

We agree that using only one example of a Class A and Class B GPCR could be seen as limiting. To strengthen the generalizability of our findings, we repeated the assays using additional receptors—D1R (Class A) and PTHR (Class B)—and observed similar Gas trafficking patterns, supporting our initial conclusions (see Fig. 5C and section “Gas endosomal redistribution induced by other class A and B GPCRs and endogenous GPCR”, page 6-7 of the revised manuscript). Regarding the term “rapid,” we acknowledge that in the context of cellular signaling, such as cAMP second messenger, minutes may not be considered fast. However, in the context of intracellular trafficking, particularly endocytosis to late or recycling endosomes, a timescale of around two minutes is generally considered rapid. For these reasons, we would like to keep the current title.

Comments:

1) The description that there is a fair amount of G α s in the cytoplasm (Line 97-99) does not appear consistent with prior studies describing the distribution of G $\beta\gamma$ subunits utilizing CRISPR-mediated gene editing (Cho et al., 2022 PMID: 35271311 and Jang et al., 2024 PMID: 39514269). I have a suspicion that this cytoplasm heavy distribution is secondary to overexpression as there are previous reports suggesting inability for free G α or G $\beta\gamma$ subunits to efficiently traffic to plasma membrane (Wedegaertner, 2012 PMID: 23161140). There is also discrepancy in relative PM to cytoplasm ratio for 0 min Gas-GFP in figure 2A and 2D compared to that of Figure 1. I would recommend doing control experiments looking at relationship between expression of Gas-GFP/Rluc and spatial distribution. Easiest and least-descriptive experiment would be to do bystander BRET for Gas-Rluc. Total luminescence can be counted for overall expression amount. Moreover, there should be a quantitative assessment of Gas -GFP distribution, perhaps by line analysis.

We agree with the reviewer that the apparent cytoplasmic distribution of Gas observed in our study is likely influenced by overexpression, as reported in earlier literature. While CRISPR-based studies examining G $\beta\gamma$ subunits at endogenous levels (Cho et al., 2022; Jang et al., 2024) report predominant plasma membrane localization, prior microscopy and fractionation studies have shown that Gas can be found both at the plasma membrane and in the cytoplasm at steady state (PMID: 8856666; 12135370). To address the impact of expression levels, we systematically varied transfection conditions and confirmed that increased G α s expression correlates with augmented cytoplasmic localization, while lower expression favors plasma membrane enrichment. Nonetheless, we observed notable cell-to-cell heterogeneity in Gas-GFP distribution even at low expression, likely explaining discrepancies between figure panels. For all analysis, we selected cells with moderate expression and clear plasma membrane localization to best approximate physiological localization. To improve clarity and consistency, we have now updated the relevant figure panels using more carefully selected representative cells that reflect physiological localization patterns or even repeated the experiments (as described in comment #2 below). We appreciate the reviewer's suggestion to use bystander BRET or line scan analysis to assess the relationship between G α s expression and membrane proximity. However, we believe that such BRET measurements would not unambiguously distinguish between membrane and cytoplasmic distribution across different expression levels, and would require extensive optimization and additional controls. While line scan or fluorescence quantification could provide further details, our optimization experiments demonstrated that Gas localization is closely linked to expression levels, and cells with appropriate plasma membrane labeling could be reliably selected at all transfection conditions. Therefore, we feel that laborious large-scale, quantitative analyses would add limited mechanistic insight beyond what is already reported—namely, that increased expression can lead to an apparent increase in cytosolic Gas due to saturation of trafficking or membrane-targeting pathways. In the revised manuscript, we have clarified the observed heterogeneity in Gas distribution and explicitly described our criteria for selecting individual cells with appropriate expression and localization profiles (see page 16).

2) It is unclear to me why receptor expression is so punctate at baseline for V2R in Figure 1B. Can authors explain this?

We agree with the reviewer that the original images show an atypical punctate pattern of V2R at the plasma membrane, which we believe was due to the use of cell-surface antibodies labeling. This method may have inadvertently promoted receptor clustering or generated labeling artifacts that resulted in the observed punctate staining. To address this, we repeated the experiments using SNAP-tagged V2R and - β 2AR labeled with a fluorescent dye that covalently binds to receptors at the cell surface. This strategy minimizes the potential for receptor aggregation and provides a more faithful representation of receptor localization. As shown in the revised figures (Figure 1 and Supplementary Figures 3-4), this approach resolved the punctate signal and confirmed more uniform membrane localization of V2R at baseline.

3. For accurate assessment of GPCR localization, it would be important to know that the majority of receptors are indeed labeled, and we are not heavily underlabeling and missing population of receptors that we cannot observe. I suggest authors to demonstrate the efficiency of labeling tag-GPCR constructs by expression tag-GPCR-Fluorescent Tag labeled by antibody in different concentrations.

We thank the reviewer for this important comment. To address the concern regarding labeling efficiency and to ensure that no significant population of receptors was missed, we performed an experiment using a HA-SNAP-tagged V2R construct. Cells expressing this receptor were first labeled at the surface using the cell-impermeant fluorescent substrate SNAP-Surface 649 and then stimulated with ligand for 10 minutes to allow receptor internalization. Following stimulation, the cells were permeabilized and stained

with an anti-HA antibody followed by an Alexa Fluor 488–conjugated secondary antibody to label the total pool of receptors. Colocalization analysis showed that almost all vesicles were double-labeled with both fluorophores, indicating that nearly all receptors had been labeled with the SNAP dye at the plasma membrane. These results, showed in the image below, support the high efficiency of our labeling strategy and suggest that no major receptor population was missed in our imaging experiments. We were not planning to add this data to the manuscript, but we can do it if required by the reviewer.

4. Authors point that Figure 1C-1D demonstrates that G α s trafficking is dependent on activation of G α s-activating GPCRs. However, there are no control experiments showing that the ligands used for CXCR4 (representative of G β i) and AT1R (representative of G β q and G β i) are indeed functional since there are no positive controls. I personally do not believe Fig 1C and 1D are needed since it's intuitive that G α s wouldn't be affected. However, if authors choose to address this question, I recommend using GFP-mini G proteins (GFP-mG β i/mG β q) or GFP-arrestin.

As recommended by the reviewer, we have removed the data presented in Figure 1C and 1D from the revised version of the manuscript.

5. It is sub-optimal that authors conducted separate imaging experiments for G α s-Rab-GPCR colocalization. It likely would have been more flexible to use SNAPf-GPCR for flexibility in avoiding wavelength overlap

We agree with the reviewer that triple labeling of GPCRs, G α s, and Rab proteins would, in principle, be the most direct approach for co-localization analysis in a single imaging experiment. We attempted such experiments, but encountered significant technical limitations, particularly during live-cell imaging. To capture the rapid dynamics of protein trafficking, fast image acquisition in non-sequential mode was required. Under these conditions, substantial spectral cross-talk occurred between channels—even with far-red dyes such as 649—making reliable signal separation unfeasible under live imaging conditions. Moreover, performing these analyses on fixed cells was not possible as we found that fixation introduced additional artifacts: notably, the intracellular membrane localization of G α s-GFP was markedly reduced in fixed samples, likely due to fixation-induced redistribution or loss of membrane association, as reported by other groups (see, bioRxiv preprint doi:10.1101/2025.05.12.653522). Given these technical challenges, we opted for live-cell imaging without fixation, imaging GPCRs, G α s, and Rab markers separately to preserve physiological localization and ensure accurate interpretation of their subcellular distribution.

6. To address whether the size of GFP tag affected G α s trafficking (Fig S1a), authors addressed that the limitation of this approach is high level of GFP signal in cytoplasm due to frankenbody-GFP that is not bound to G α s-HA. I would recommend trying split fluorescent protein strategy with mNG2 (11) tag attached to G α s and cytoplasmic expression of mNG2 (1-10). The advantage would be that mNG2 construct is mostly fluorogenic only when they are complemented.

We thank the reviewer for this thoughtful suggestion. The use of split fluorescent proteins such as the mNG2(11)/mNG2(1–10) system is indeed an elegant strategy for reducing background fluorescence and improving signal specificity. However, it is beyond the scope of this study as the primary goal of this experiment was simply to determine whether the size of the GFP tag affects Gas trafficking. We observed that HA-tagged Gas displays trafficking dynamics entirely consistent with those of Gas-GFP following β 2AR stimulation with isoproterenol. These findings indicate that the larger GFP tag does not impair Gas trafficking. While the split fluorescent protein strategy offers clear advantages in other contexts, we believe it would not provide significant additional insight for the specific purpose of this control experiment in our study.

7. For assessment of whether G $\beta\gamma$ overexpression also affects the distribution Gas, it is unusual that the authors focused solely on G β 1 γ 1 since farnesylation of G γ 1 affects the distribution and kinetics of dimer translocation upon activation (Kankanamge et al., 2022 PMID: 36272647). Namely, farnesylated gamma subunits (1,9,10) are more readily distributed away from PM upon activation. Because the confocal data did not suggest change in distribution for this condition, I do not believe testing other gamma subunits is necessary. However, this should be addressed. If authors choose to keep the G $\beta\gamma$ overexpression data, there should be control assessing whether G β 1 γ 1 is indeed the predominant dimer forming complexes with Gas -GFP.

We agree that G β 1 γ 1 may not represent the predominant G $\beta\gamma$ dimer coupling with Gas. Furthermore, as noted by Reviewer #1, the role of G $\beta\gamma$ in regulating Gas trafficking has been primarily associated with class B GPCRs, such as the V2R, as reported by Sokrat et al., 2024 (PMID: 38972875). To address this, we performed additional experiments in HEK293 cells overexpressing V2R along with G β 1 γ 2, a G $\beta\gamma$ combination more relevant to the V2R-Gas signaling context. Upon V2R stimulation, we observed that G β 1 γ 2 overexpression did not alter Gas trafficking to vesicles as assessed by confocal microscopy. Similarly, in our BRET-based assays, G β 1 γ 2 had no significant effect on Gas recruitment to Rab-labeled endosomal compartments (Supplementary Figure 6). These results suggest that G $\beta\gamma$ overexpression, at least in this context, does not influence Gas trafficking.

8. The fact that there is affinity (perhaps in micromolar scale) between rGFP and RLucII should be mentioned (Line 163-165) per Molinari et al., 2008 (PMID: 17868039) and Namkung 2016 (PMID: 27397672). While the affinity is strong enough to bias the system towards intracellular redistribution is questionable, it should nevertheless be mentioned. Furthermore, the affinity is even more unclear in the context of cytoplasm to membrane or membrane to membrane, that is 3D to 2D and 2D to 2D interactions, respectively

This is now mentioned in the text (see page 5): “Although RLucII and rGFP can self-associate with moderate (micromolar) affinity (Molinari et al., 2008, PMID: 17868039; Namkung et al., 2016, PMID: 27397672), this intrinsic interaction facilitates efficient BRET transfer and has been leveraged to generate sensitive trafficking sensors, such as β -arrestins to early endosome (Namkung et al., 2016). Therefore, although this intrinsic interaction may contribute to basal BRET signals, changes in Gas recruitment to endosomes are expected to increase BRET signals regardless.”

9. Please show baseline BRET for steady state experiments (Figure 5D). While the baseline number is impacted by conformation and not entirely reliable for quantitative purpose, clear differences especially in comparison between different GPCR overexpression may provide additional information. Plus, also useful for assessing validity of the experiments.

In the original version of the manuscript, baseline BRET values were shown for all BRET experiments. However, in response to Reviewer #1’s request, we revised the figures to display averaged normalized BRET changes (Δ BRET) with corresponding SEM values. That said, we acknowledge the value of including baseline BRET signals for comparison. To address this, we now provide the raw BRET2 ratio values as supplementary data (Supplementary Figures 2 and 7), which should aid in the interpretation and validation of the experiments.

10. It would be ideal to quantify kinetics for Figure 8B

The kinetics of Gas dissociation from the plasma membrane (Fig. 6E in the revised version of the manuscript) and its translocation to Rab7-positive endosomes (Fig. 6H in the revised version of the manuscript) have now been quantified by measuring the half-time ($T_{1/2}$) of these processes. These $T_{1/2}$

values are provided within the manuscript (page 8), offering a direct comparison of the kinetic profiles for each process.

11. What accounts for baseline BRET difference (much lower in fig. S7 around 0.24 than fig. 8B 0.4)?

The difference in baseline BRET values between previous Figure S7 and Figure 8B is due to the use of different RLucII substrates. Specifically, Prolume Purple was used in Figure S7, whereas coelenterazine 400A was used in Figure 8B. For reasons that remain unclear, coelenterazine 400A produced very poor BRET signals with the myristoylated Gas67-RLucII construct, making those results difficult to interpret. In contrast, using Prolume Purple resolved this issue and yielded clearer, more reliable signals. As a result, Prolume Purple was used for the experiments (Gas -WT and Gas-Myr) in Figure S7B (Fig. 8B in the revised version of the manuscript), accounting for the lower baseline BRET values observed in that figure. It is worth noting that Prolume Purple has been reported to produce lower absolute BRET signals compared to traditional substrates, due to its lower intrinsic brightness, faster signal decay, and, in some configurations, less efficient spectral overlap with standard BRET acceptors. These properties contribute to its lower photon yield and explain the lower baseline BRET values (PMID: 31131059).

12. Period at the end of paragraph Line 184.

This has been corrected in the revised manuscript.

13. Was the Dyn K44A experiment done in the context of using bystander BRET? If so, did it line up with the confocal findings?

We have now confirmed the DynK44A results from Figure 7 using bystander BRET, as shown in the newly added Figure 7B-D. Consistent with our confocal imaging data, bystander BRET measurements revealed that Gas trafficking was not impaired by DynK44A overexpression—both in terms of its dissociation from the plasma membrane (using rGFP-CAAX) and its recruitment to late endosomes (using rGFP-Rab7). In contrast, β 2AR transport to early endosomes was largely inhibited. These findings further support the conclusion that Gas internalization occurs independently of dynamin-mediated endocytosis.

14. In Line 365-367, authors claim that the Gas endosomal translocation is independent of β -arrestin recruitment, but this was not experimentally addressed by authors. Please either conduct the appropriate experiments or cite other studies that have suggested it (Kwon et al., 2022 PMID: 36289326).

Our original conclusion that Gas endosomal translocation is independent of β -arrestin recruitment was based on experiments with dobutamine, a ligand known not to recruit β -arrestins (as confirmed by its inability to induce β 2AR internalization) but still capable of promoting Gas trafficking upon β 2AR activation. While we did not directly ablate β -arrestin function, these findings support the notion that, at least in the context of β 2AR, Gas translocation can occur independently of β -arrestin. To further strengthen this point, we have now cited other studies (Kwon et al., 2022; Texeira et al 2025; Wysolmerski, et al 2025), which provides additional experimental evidence for β -arrestin-independent mechanisms of Gas endosomal trafficking. This reference and clarification have been added in the Discussion (page 13), as suggested by the reviewer.

Please find enclosed our revised version of manuscript ID COMMSBIO-24-8266A by Laniel et al., initially entitled "*Stimulation of class A and class B GPCRs leads to rapid translocation of Gas to late and slow recycling endosome*". We wish to thank the reviewers for their helpful suggestions, which have improved the manuscript. We believe that we have adequately addressed all the issues that were raised. Below, you will find our detailed responses to the reviewers' comments.

Reviewers' comments:

Reviewer #1 (Remarks to the Author):

Laniel et al. have done an excellent job addressing most of the concerns previously raised. As a result, the findings presented in the manuscript are now significantly more robust and convincing. Below are a few remaining concerns that should be readily addressable:

1- I remain concerned about the lack of effect of G $\beta\gamma$ on Gas trafficking. I wonder whether the use of Gas119-RLucII in the present manuscript (rather than Gas67-RLucII as used by Sokrat et al.) masks the impact of G $\beta\gamma$ overexpression on Gas trafficking. Could the authors test the effect of G $\beta\gamma$ overexpression on Gas67-RLucII trafficking? FYVE and Rab5 are both well-established markers of early endosomes making the Gas119-RLucII construct the most plausible explanation for the discrepancy between the present study and that of Sokrat et al. This experiment would be crucial to ensure the results are not artefactual (especially given that the findings appear to contradict a pivotal concept recently published by the Bouvier group).

We thank the reviewer for this comment and the opportunity to clarify. All of our assays were performed with Gas119-RLucII, selected after a direct comparison with Gas67-RLucII. Both biosensors displayed the same endosomal trafficking profile; however, Gas119 provided a stronger and more reliable BRET signal, which is why we used it consistently throughout the manuscript. Importantly, this biosensor (also referred to as Gas117-RLucII) has been widely validated and employed by several groups, including those of R. Lefkowitz, B. Kobilka, and M. Bouvier (PMID: 27499021, 32259053, 30327561). As we note in the Discussion (page 13), differences between biosensors may account for variations across studies, as is often reported with BRET-based approaches. We do not challenge previous findings by the Bouvier group but rather acknowledge that methodological differences are a plausible explanation for the discrepancy. We also wish to highlight that the data on G $\beta\gamma$ overexpression were included only in the supplementary figures, as they are not central to our main conclusions and that repeating the entire set of experiments with Gas67 would not change the conclusions or the main message of our manuscript. Such work is interesting but would primarily constitute a comparative study of biosensor performance, which falls outside the scope of the present paper.

2- The new panels 6F and 6I are very interesting as they demonstrate that Gas translocation from the plasma membrane to late endosomes is concentration-dependent. While the authors state that similar potencies are observed for each ligand, there are clearly noticeable differences that should not be overlooked. In particular, the potency of dobutamine appears significantly lower than that of isoproterenol, salbutamol, and epinephrine. This should be acknowledged and if possible briefly discussed with a possible explanation.

We thank the reviewer for this insightful comment. Our intention was to highlight that, for each ligand, the potency values were highly similar when comparing the plasma membrane (CAAX) and late endosome (Rab7) compartments, rather than to imply equivalence across all ligands. We agree with the reviewer that dobutamine displays a noticeably lower potency compared to the other ligands. We have now clarified these points in the Results section (page 9) and included, as possible explanations, that the lower potency of dobutamine is consistent with previous reports describing its weaker ability to stimulate β_2 AR-mediated Gas recruitment and cAMP production (PMID: 35732075).

Reviewer #2 (Remarks to the Author):

Laniel & al aimed at investigated the spatio-temporal distribution of GalphaS upon two prototypical Class A & class B GPCRs, the beta2-adrenergic receptor and the arginine-vasopressin receptor type 2, respectively.

Authors added new data that clearly strenghten and support the observation that GalphaS is redistributed intracellularly with an apparent enrichment on Rab7 and Rab11 positives endosomes and extended these observation with two additional representative GPCRs of class A & class B. This work further add to better understand how G proteins (here GalphaS) are regulated beyong activation and dissociation from Gbetagamma. Nonetheless, further work would be needed to understand a possible function of GalphaS in the different rab7 and Rab11-endosomes. In my opinion, Laniel & al have answered most criticis and comments raised upon the initial round of review and in my opinion, the revised manuscript is convincing and is now compatible with publication.

We sincerely thank the reviewer for their thorough evaluation and constructive feedback throughout the review process. We are pleased that the additional data and clarifications have strengthened the manuscript and that the revised version is now found convincing.

Reviewer #3 (Remarks to the Author):

Major Impression 2:

While I agree that these are indeed the differences in experiment design between the current study and prior studies, I am not sure whether the current study fundamentally changes our understanding of activated Gas that we suspect do sample different endosomal compartments. And without monitoring endogenous G α subunits (which no group has published at this point of writing), it is difficult to say whether overexpression-based methods make substantial progress on this issue. The findings would be limited to the fact that engineered tools are used to postulate how the endogenous protein would behave and differences noted here are directly compared to prior tools. I am not sure how using ebBRET over BRET1 was necessary to demonstrate the findings as well.

We fully agree with the reviewer that monitoring endogenous G α subunits trafficking, as well as assessing their activity across different endosomal compartments, represents an essential next step in the field.

Major Impression 3:

While the timing of what is "rapid" is relative to what process we are comparing to, I still believe we cannot generalize the current studies to Class A and Class B GPCRs based on two receptors for each category. There is no specific number that meets the cutoff of what makes a certain study generalizable to the entire class but studying two receptors per class is on the smaller scale, especially considering BRET-based assays are pretty scalable. While I do not think it is fundamental for the authors to study more receptors to prove their points, it would be misleading to state the title as it is. Please consider removing the context of Class A and Class B GPCRs and rather emphasize what makes Gas signaling different.

We appreciate the reviewer's thoughtful concern. Our intention was not to generalize that all receptors in Class A and B behave identically, but rather to emphasize that the representative receptors we examined from each class showed a similar pattern of Gas translocation. The reference to Class A and B GPCRs is important, as these classes are known to differ in their levels of endosomal signaling and intracellular trafficking. Our study provides, for the first time, a direct comparison of Class A and Class B GPCRs in the context of Gas translocation, using two well-characterized receptors from each class. While we acknowledge that examining a larger number

of receptors would increase generalizability, our findings are consistent across the selected representatives and provide novel mechanistic insights. We therefore believe that the current title appropriately reflects the scope and novelty of the study and would like to keep it.

Comment 1:

It is recommended to explicitly state that the apparent cytoplasmic distribution of Gas is likely due to overexpression as there is currently no available data of endogenous Gas distribution in live cells. The cell-to-cell variability should be mentioned in results/discussion, not just methods as this is an important point for interpretation. Transfection at low level helps but does not resolve the issue because either/or the amount of plasmid delivered per cell and efficiency of expression are variable processes that are difficult to be titrated.

We have now incorporated a statement in the Results section (page 3) explicitly noting that the apparent cytoplasmic distribution of Gas is likely attributable to overexpression, and we highlight the cell-to-cell variability observed to ensure this limitation is clearly acknowledged in the interpretation of our findings.

Comments 2-5: Authors have addressed this question.

Comment 6: While authors do not have to conduct the suggested study, I believe it is important to mention that it should not be assumed that Gas-HA has similar trafficking as wildtype Gas just because it has a smaller tag.

We now acknowledge in the Results section (page 4) that although these findings suggest that tag size does not overtly affect Gas trafficking, it remains possible that even small tags introduce subtle effects on protein behavior; therefore, these constructs may not fully recapitulate wild-type Gas trafficking.

Comments 7-14: Authors have addressed this question.

Additional correction:

- Please include subscripts for receptors (2 for β 2AR, V2R) and G proteins (s for Gs, Gas)
These modifications have been performed.
- Line 15: “Heterotrimeric Gas is classically known for mediating G protein-coupled receptor (GPCR)”
Since Gas is monomeric, would say “Gas is classically known...”. Or could say heterotrimeric Gs
This modification has been performed.